# A distributed nanocluster based multi-agent evolutionary network

Liying Xu[1], Jiadi Zhu[1], Bing Chen[2], Zhen Yang[1], Keqin Liu[1], Bingjie Dang[1], Teng Zhang[1], Yuchao Yang ®[1,3,4,5] ✉ & Ru Huang ®[1,3,4] ✉

As an important approach of distributed artificial intelligence, multi-agent system provides an efficient way to solve large-scale computational problems through high-parallelism processing with nonlinear interactions between the agents. However, the huge capacity and complex distribution of the individual agents make it difficult for efficient hardware construction. Here, we propose and demonstrate a multi-agent hardware system that deploys distributed Ag nanoclusters as physical agents and their electrochemical dissolution, growth and evolution dynamics under electric field for high-parallelism exploration of the solution space. The collaboration and competition between the Ag nanoclusters allow information to be effectively expressed and processed, which therefore replaces cumbrous exhaustive operations with self-organization of Ag physical network based on the positive feedback of information interaction, leading to significantly reduced computational complexity. The proposed multi-agent network can be scaled up with parallel and serial integration structures, and demonstrates efficient solution of graph and optimization problems. An artificial potential field with superimposed attractive/repulsive components and varied ion velocity is realized, showing gradient descent route planning with self-adaptive obstacle avoidance. This multi-agent network is expected to serve as a physics-empowered parallel computing hardware.

A complex system, including the world, can be constructed by a large network of relatively simple components with evolution capabilities. The self-organization of the individuals following simple rules can lead to complex behaviors of the network system[1]. For instance, the swarm that simple biological individuals constitute shows collective intelligence capable of solving complex computational problems[2–8]. Swarm intelligence-inspired algorithms such as genetic algorithm[9], ant colony algorithm[10], and particle swarm optimization algorithm[11] have played an important role in optimization and distributed artificial intelligence (AI).

In fact, all the complexity comes from simplicity[3]. The swarm intelligence essentially arises from the emergence and self-organization characteristics[2,12,13]. The evolution of simple individuals through nonlinear interactions under de-centralized collaborative control[14] makes the whole show new structures or functions that the individuals do not possess, that is, "the whole is greater than the sum of its parts". As a considerable branch of distributed AI, multi-agent system inspired by swarm intelligence can deal with complex tasks that far exceed the capability of single agents, which is based upon the interactions between the agents as well as that between the agents and the

[1]National Key Laboratory of Science and Technology on Micro/Nano Fabrication, School of Integrated Circuits, Peking University, 100871 Beijing, China. [2]School of Micro-Nano Electronics, Zhejiang University, 310058 Hangzhou, Zhejiang, China. [3]Center for Brain Inspired Chips, Institute for Artificial Intelligence, Peking University, 100871 Beijing, China. [4]Center for Brain Inspired Intelligence, Chinese Institute for Brain Research (CIBR), Beijing, 102206 Beijing, China. [5]Beijing Academy of Artificial Intelligence, 100084 Beijing, China. ✉e-mail: yuchaoyang@pku.edu.cn; ruhuang@pku.edu.cn

environment, showing stupendous potential in diverse applications, such as swarm robots[15,16], multivehicle coordination[17], and machine learning[18,19]. Despite the significance of multi-agent systems, the huge capacity and initially random but subsequently complex distribution of individuals make it difficult for hardware implementation, and hence previous studies are mainly focused on related algorithms. The development of multi-agent evolutionary hardware systems will be key to future construction of large-scale, energy-efficient, self-adaptive, and robust AI systems.

In recent years, physical networks with nanoarchitecture composed of nanowires[20–24], nanoparticles[25–27] or random dopants[28] have been exploited to efficiently implement complex computational tasks in material based on the nonlinear interactions of the individuals. Gimzewski et al. designed atomic switch networks (ASN) composed of multiple overlapping Ag nanowire junctions. The distributed spatiotemporal dynamics of ASN shows great potential for the efficient implementation of reservoir computing[20,21]. Brown et al. studied the avalanches, and self-organized criticality originated from spatiotemporal correlations in percolating nanoparticle network. The statistical distributions of an avalanche in the percolating nanoparticle network exhibit qualitative and quantitative similarity to those measured in the cortex, which provides a novel architecture for efficient brain-like computing[26,27]. van der Wiel et al. exploited the nonlinearity and tunability of hopping conduction in a silicon-based network of boron dopant atoms, enabling efficient implementation of machine learning tasks such as classification[28]. The rich and complex dynamics of these physical networks have shown great potential in unconventional computing with high energy efficiency.

Here, we report a nanoscale, solid-state multi-agent evolutionary network (MAEN) based on the self-organization of distributed Ag nanoclusters. Different from the existing physical networks with immobile elements, here, the Ag nanoclusters in the MAEN system as de-centralized agents exhibit spontaneous dynamic evolutions via field-driven ion migrations and electrochemical reactions. The positive feedback by the alignment of Ag nanoclusters leads to cooperation and competition between the agents, and the resultant connectivity pattern conforms to the principle of optimization. The self-organized evolution of Ag nanoclusters with positive feedback has commonalities in principle with swarm intelligence, such as the foraging process of ant colony. The kinetic factors in this process, including electric field and ion mobility, provide effective means to the modulation of evolution dynamics. Two types of basic modulation units, including distance modulation and voltage modulation, are built accordingly and can effectively represent the weights of edges in a graph. We further propose and experimentally demonstrate parallel and serial integration schemes of basic modulation units, which indicate the scaling potential of the MAEN and are successfully applied to the solution of varied graph problems. An artificial potential field with superimposed attractive and repulsive components and thus varied ion velocity is experimentally realized, showing gradient descent route planning with obstacle avoidance. The distributed Ag nanocluster-based MAEN with evolutionary capability and high parallelism, therefore, provides an efficient route toward realizing distributed AI hardware.

## Results

### Operation principle

In complexity science, the phenomenon of swarm intelligence can be described by "emergence", originating from a positive feedback amplification mechanism based on local interactions between the individuals[29,30]. The ant colony foraging process is a typical example embodying the emergence behavior of complex systems, as schematically depicted in Fig. 1a. In the initial food searching stage, the ants do not know the location of food and tend to randomly search

the entire space. A chemical substance called "pheromones"[10] will be released to the surrounding environment once the ants find food. The pheromones can guide other ants to find the food, and meanwhile the concentration of pheromones will dissipate over time. If there exists a path that is shorter than the others, the time it takes for passing it will be shorter, and thus the concentration of pheromones thereof will be higher. It naturally attracts more ants to this path, which in turn generates more pheromones. This forms positive feedback and ensures that the ants eventually converge to the shortest path to transport food, as schematically shown in Fig. 1b. Inspired by the foraging law, ant colony algorithms are widely used to solve optimization problems, such as the shortest path problem.

Although it is difficult to artificially predict and manipulate the emergence itself, it can be induced by designing the conditions that prompt emergence to occur. Here, we propose a MAEN system based on the self-organized evolution of Ag nanoclusters under an electric field. The preparation of the devices is described in Experimental Section and Supplementary Fig. 1. Self-assembled polyethylene oxide (PEO) with high $Ag^+$ ion mobility was selected as the solid polymer electrolyte[31,32]. Figure 1c shows scanning electron microscopy (SEM) image of the initial morphology of a two-terminal MAEN device with ~200 nm gap. It can be seen that a large number of Ag nanoclusters with a diameter of ~10 nm are randomly distributed and incorporated in the device (Supplementary Fig. 2), which can be regarded as bipolar electrodes upon application of an electric field[33,34]. As a result, the self-organized evolution of such clusters as distributed agents can be achieved by a sequence of ionization, ion migration, and reduction processes of individual Ag atoms, leading to effective cluster displacements along electric field and thus alignment of them (Fig. 1d). The aggregation process of nanoclusters in turn reduces their distances and enhances the local electric field, forming a positive feedback mechanism. As a result, the randomly distributed Ag clusters eventually self-organize into a conductive filament connecting the two terminals, as shown in Fig. 1d and also verified by atomic force microscopy (AFM) analysis (see Supplementary Fig. 3). A significant increase of current is observed simultaneously (Supplementary Fig. 4, Supporting Information), suggesting that an electrical connection is established between the two terminals.

The physically evolving process is further studied using kinetic Monte Carlo (MC) simulations (see Experimental Section and Supplementary Fig. 5), as shown in Fig. 1e (upper panels), along with corresponding electric field distribution (bottom panels). Initiating from the original state with randomly distributed Ag clusters (Fig. 1e1), the most visible evolution of Ag clusters occurs at places with the highest electric field intensity due to the field-assisted ion transport[35]. The conductive filament growth mainly proceeds along the shortest path between the terminals, in agreement with experimental results (Fig. 1d). The morphology of conductive filament can be tuned by the compliance current[31] (Supplementary Fig. 6), and pinched hysteresis loops were observed during $I-V$ measurements and verified the memristive effect[36–41] of the two-terminal MAEN devices (Supplementary Figs. 7–9). The threshold switching characteristics indicate the device spontaneously relaxed back to the off state after removing the voltage bias, which is exactly due to the filament dissolution facilitated by interfacial energy minimization between Ag nanoclusters and dielectrics[42,43]. The MAEN system with nonlinear dynamics and fading memory property holds potential in efficient implementation of neuromorphic computational tasks, such as reservoir computing[20,21,24]. Systematic studies on a large number of two-terminal MAEN devices reveal that the evolution kinetics can be regulated by device configurations and stimulus parameters, i.e., the hopping of $Ag^+$ ions is a thermally activated process[44] and the activation energy $U'$ during ion transport will be lowered by a term linearly dependent on the electric field to the first

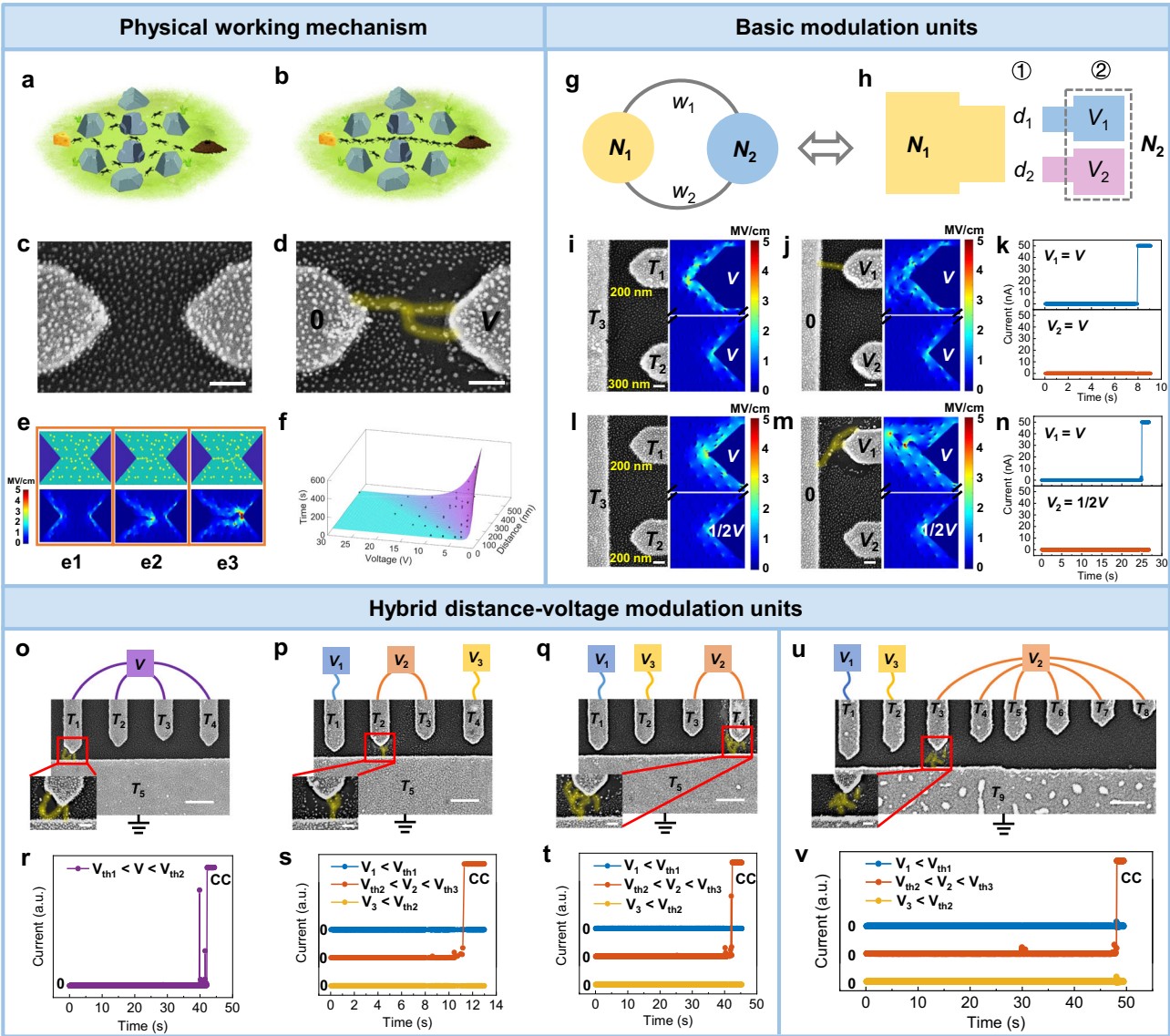

**Fig. 1 | Physical operation principle and construction of modulation units.**
Schematic diagram of the ant distribution in **a**, the early foraging period and **b**, the later foraging period. The originally randomly distributed ants finally converge to the shortest path to transport food. SEM image of the two-terminal device with ~200 nm gap (**c**), and after switching (**d**). $V = 25$ V. Scale bar: 100 nm. **e** Monte Carlo simulation of the Ag atom/cluster evolving (upper) and corresponding electric field distributions (bottom). **f** Forming time as a function of gap distance and applied voltage. **g** Graph structure composed of two nodes connected by two edges. **h** Schematic diagram of device structure and mapping schemes corresponding to (**g**). **i** SEM image (left) and initial electric field distribution (right) of distance modulation unit. **j** Corresponding state after switching. Scale bar: 100 nm. **k** Time-dependent current measurement from $T_1$ (blue) and $T_2$ (orange). $V = 10$ V. **l** SEM image (left) and initial electric field distribution (right) of voltage modulation unit.

**m** Corresponding state after switching. Scale bar: 100 nm. **n** Time-dependent current measurement from $T_1$ (blue) and $T_2$ (orange). $V = 30$ V. **o–q** SEM images of connectivity pattern in a five-terminal device ($V = 20$ V, $V_1 = 10$ V, $V_2 = 25$ V, $V_3 = 15$ V). The designed gap distance of each device was 100, 200, 300, and 200 nm from left to right, respectively. Scale bar: 500 nm. The insets show the magnified filaments, scale bar: 100 nm. **r–t** Time-dependent current measurements corresponding to **o–q**. CC: 50 nA. The blue, orange, and yellow curves in **s** and **t** were from $V_1$, $V_2$, and $V_3$ electrodes, respectively. **u** SEM image of connectivity pattern in a nine-terminal device ($V_1 = 10$ V, $V_2 = 25$ V, $V_3 = 15$ V). The designed gap distance from left to right is 100, 200, 200, 300, 400, 500, 600, and 700 nm, respectively. Scale bar: 500 nm. The insets show the magnified filaments, scale bar: 100 nm. **v** Time-dependent current measurement from the $V_1$ electrode (blue), $V_2$ electrode (orange), and $V_3$ electrode (yellow) corresponding to **u**. CC: 10 nA.

order[35], i.e., $U' = U − \alpha E$. The forming time of filament $t$ is thus determined by:

$$t = \frac{1}{f} \cdot e^{U'/k_B T} = \frac{1}{f} \cdot e^{U/k_B T} \cdot e^{-\alpha V/k_B T d} \quad (1)$$

where $f$ is the attempt frequency, $k_B$ is the Boltzmann's constant, and $T$ is the absolute temperature. Figure 1f shows that the experimental data obtained from multiple devices with different gap distances and applied voltage biases can be fitted into a curved surface corresponding to Eq. (1). The forming time $t$ decreases exponentially with

applied voltage bias $V$ and increases with gap distance $d$, which indicates that the electric field ($E = V/d$) driven Ag cation transport is the rate-limiting process.

Notably, the self-organized evolution of Ag nanoclusters with positive feedback is inherently analogous to the foraging process of the ant colony. The initial random distribution of Ag nanoclusters (Fig. 1c) corresponds to the unorganized states of ants in the early foraging stage (Fig. 1a). Movable Ag$^+$ ions under electric field and the conductive filament(s) are analogous to the ants and the pheromones released, respectively. The positive feedback of filament growth can well map the feedback mechanism of pheromone release. While the

ants find the shortest path with the aid of pheromones (Fig. 1b), the Ag nanoclusters spontaneously align into a filament along the path with the shortest equivalent distance (Fig. 1d, e). The time complexity needed for self-organization of Ag nanoclusters is only $O(1)$.

## Basic and hybrid modulation units

The Ag nanocluster-based MAEN device with two terminals can be easily extended to multiterminal MAEN. Upon multiple input signals, the positive feedback accelerates filament growth along the optimal path through the cooperative interactions between Ag clusters on the same connection path, while there exist competitive interactions between Ag clusters on different connection paths. Complex functions that individual Ag nanoclusters do not possess can be realized by the whole composed of individuals based on the collective self-organized evolution of Ag clusters.

The MAEN devices with two and multiple terminals can be abstracted into graphs in general. Figure 1g illustrates a simple graph, where two edges with different weights ($w_1$ and $w_2$) are connected between the nodes $N_1$ and $N_2$. The problem to be solved is to find the edge with the smallest weight between $N_1$ and $N_2$. Here, the electrodes of MAEN are used to represent the nodes of the graph. Since the growth kinetics of conductive filament is regulated by electric field intensity, both the gap distance and voltage bias can be used to represent the weights of edges. Therefore, the distance modulation unit and voltage modulation unit can be constructed as two basic building blocks, as shown in Fig. 1h. For distance modulation unit (①, Fig. 1h), the distances of gaps (e.g., $d_1$ and $d_2$) can be used to represent edges with different weights (e.g., $w_1$ and $w_2$), while the applied voltage to the gaps is kept identical ($V_1 = V_2$). For voltage modulation unit (②, Fig. 1h), the applied voltage biases (e.g., $V_1$ and $V_2$) are used to represent edges with different weights, while the distance of gaps keeps unchanged ($d_1 = d_2$).

We assume that the upper edge has a smaller weight than the lower edge ($w_1 < w_2$) in Fig. 1g. The corresponding distance modulation unit is thus shown in Fig. 1i, along with electric field distribution in the initial state using kinetic Monte Carlo simulation. The electric field intensity between terminals $T_1$ and $T_3$ is higher under the same voltage bias ($V_1 = V_2$), due to the shorter gap distance compared with that between $T_2$ and $T_3$ ($d_1 < d_2$). The originally small difference between the gap distances is amplified by the field-driven alignment and nonlinear interaction of the Ag clusters, and hence the initial growth of filament further amplifies the difference in gap distance. Such positive feedback ensures Ag clusters self-organize into a complete filament connecting terminals $T_1$ and $T_3$, as verified in Fig. 1j, where a significant increase of current from $T_1$ is detected ($t \approx 8$ s) and confirms the connection (Fig. 1k). The same graph can be represented by a voltage modulation unit as well (Fig. 1l). From the kinetic MC simulation, one can see that higher applied voltage bias between $T_1$ and $T_3$ ($V_1 > V_2$) leads to enhanced electric field therein. As expected, the filament alignment initiates between $T_1$ and $T_3$ and proceeds due to the positive feedback mechanism, once again leading to filament formation between $T_1$ and $T_3$ (Fig. 1m, n). Both the two basic modulation units find the edge with smaller weight, i.e., $N_1 \overset{w_1}{\leftrightarrow} N_2$ (Fig. 1g), showing their equivalence. Detailed Ag cluster evolution processes of the two basic modulation units shown by kinetic MC simulations were given in Supplementary Figs. 10 and 11, where interchanging the terminals $T_1$ and $T_2$ in the unit does not affect the results (Supplementary Fig. 12).

The distance and voltage modulation units as two fundamental building blocks of MAEN system have their own advantages and disadvantages in terms of mapping and solution of the problems. For the distance modulation unit, the graph information is directly mapped with the device structure, which facilitates efficient solution of problems and saves overhead in electrode area, but leads to poor reconfigurability since the weights are represented in hardware. On the contrary, for the voltage modulation unit, the weights are represented

by the voltage biases applied to the electrode terminals, so that the same device structure can be used to represent different graph problems, hence leading to higher reconfigurability. To meet the requirements in different application scenarios, an optimal modulation scheme can be designed to improve the efficiency and flexibility for problem solution by tuning gap distance $d$ and voltage bias $V$ separately or collectively based on the respective characteristics of the two modulation units. Under the hybrid distance–voltage modulation scheme, different input information can be directly integrated in the MAEN, thus enhancing the computing efficiency. As an example, multiterminal devices with the designed size of 100, 200, 300, and 200 nm gaps (left to right) were fabricated to demonstrate hybrid distance–voltage modulations (Fig. 1o–q and Supplementary Fig. 13a). The gap distance and applied voltage will collapse into the electric field and jointly modulate the connectivity pattern. Figure 1o–q shows the connectivity patterns of conductive filament under three different voltage bias schemes, with corresponding $I$–$t$ curves displayed in Fig. 1r–t. $V_{th1}$, $V_{th2}$ and $V_{th3}$ were defined as the threshold voltages corresponding to 100, 200, and 300 nm gaps for obvious current increase within 1 min, respectively. In response to external electrical stimulations, Ag nanoclusters adaptively evolved to form different connectivity patterns depending on the combined input schemes (Fig. 1o–t), and the final connection path represents the optimal decision after comprehensively considering the cost (distance) and reward (voltage). The consistency of results is verified in multiple devices (Supplementary Fig. 13b, c). The number of terminals in the basic modulation unit can be flexibly extended. Figure 1u shows the MAEN device with 9 terminals and hybrid distance–voltage modulations, where the final connectivity pattern once again corresponds to the optimal solution (Fig. 1u, v).

## Integration and scaling schemes

To solve larger-scale problems, the fundamental units of the MAEN have to be scalable. We further propose and demonstrate two integration schemes of the above basic modulation units, namely a parallel structure (Fig. 2a–c) and a serial structure (Fig. 2d–f). Figure 2a illustrates a graph where the edges between $N_1$ and $N_2$ exist in two different space dimensions, represented by two mutually perpendicular planes (i.e., $x = 0$, $y = 0$), and each plane contains edges with different weights. The shortest path between $N_1$ and $N_2$ can therefore be found by searching the shortest path in the respective planes in parallel. Figure 2b shows two basic modulation units embodying a parallel integration structure, where the edge with the smallest weight (i.e., $N_1 \overset{1}{\leftrightarrow} N_2$, Fig. 2a) could be found after searching the two different space dimensions, and Fig. 2c shows the corresponding time-dependent measurement. In contrast, Fig. 2d illustrates a graph structure where an intermediate node $N_3$ exists between nodes $N_1$ and $N_2$, with the edges and corresponding weights marked accordingly. To find the path with the smallest weight between nodes $N_1$ and $N_2$, a serial structure is required. Figure 2e shows two basic modulation units embodying a serial integration structure. In this case, overall two filaments can be connected in series when a voltage bias $V$ is applied to $N_1$ and $N_2$, with one filament between terminals $T_1$ and $T_3$ and the other filament between terminals $T_4$ and $T_6$ (Fig. 2e, f). This thereby implies the selection of two combined edges with overall weight of 2 in total using the MAEN system, i.e., $N_1 \overset{1}{\leftrightarrow} N_3 \overset{1}{\leftrightarrow} N_2$ (Fig. 2d) and the reproducibility of the integration structures is verified experimentally (Supplementary Fig. 14). The above results unambiguously demonstrated the scaling-up potential of the MAEN system based on the parallelly and serially integrated modulation units. Notably, the existences of a limited set of standard modules (Fig. 1) and proper integration schemes (Fig. 2) are highly desirable for applications.

It is worthwhile noting that the physiochemical processes involved and the inhomogeneity of cluster distribution have introduced intrinsic stochasticity into the Ag nanoclusters-based multi-

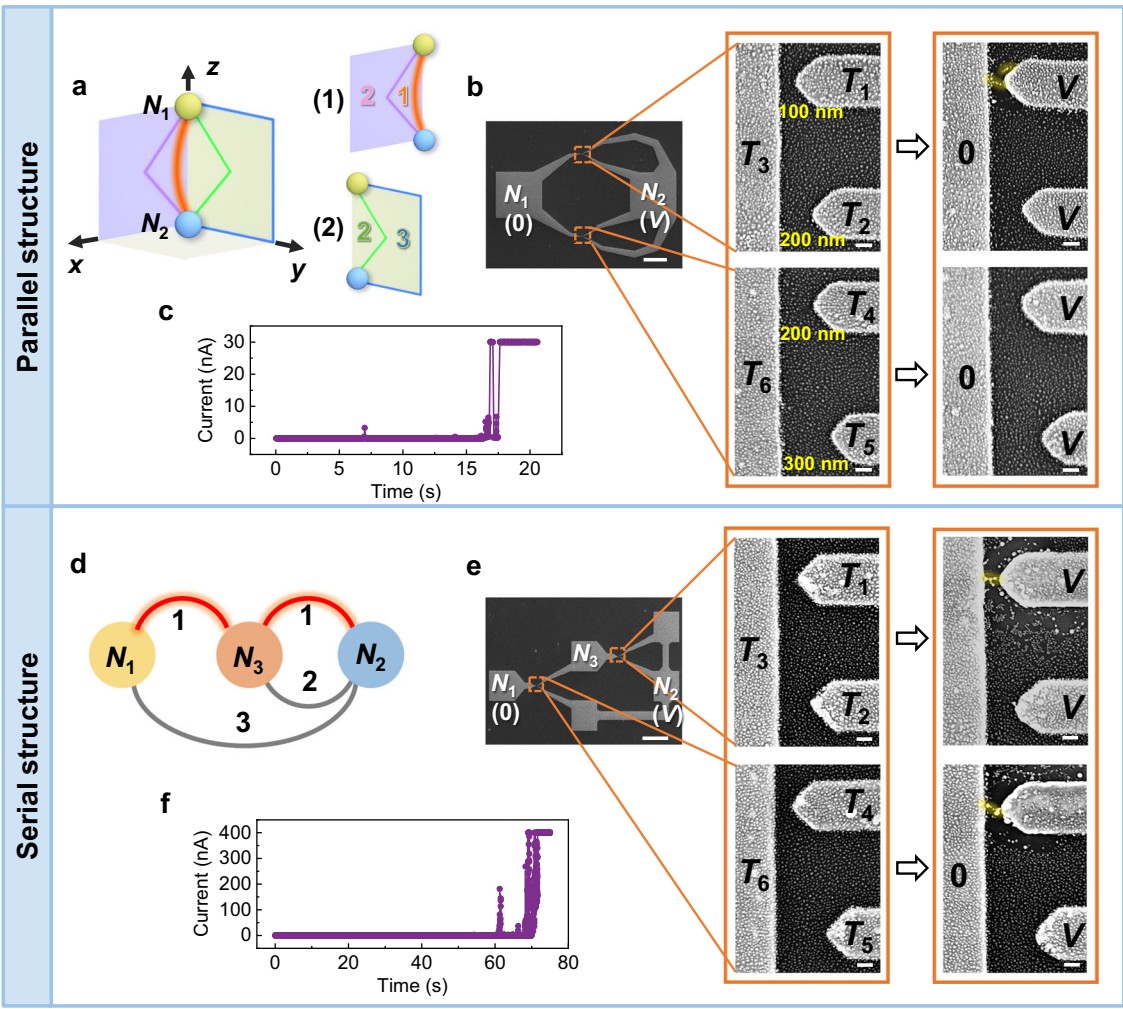

**Fig. 2 | Integration schemes of the modulation units. a** Schematic illustration of a 3D graph structure. Four edges with different weights between the nodes $N_1$ and $N_2$ exist in the two different space dimensions represented by two mutually perpendicular planes (i.e., $x = 0$, $y = 0$), where the edge with the weight of 1 (orange color) and the edge with the weight of 2 (pink color) exist in the (1) plane $y = 0$ while the edge with the weight of 2 (green color) and the edge with the weight of 3 (blue color) exist in the (2) plane $x = 0$. **b** SEM images of a parallel structure composed of two basic modulation units mapping with the graph structure in **a**: global configuration and voltage bias scheme (left panel, scale bar: 50 μm); initial morphology of each unit (middle panel, scale bar: 100 nm); connectivity pattern of conductive filament after memristive switching (right panel, scale bar: 100 nm). **c** Corresponding time-dependent current measurement from the $N_2$ electrode

under the voltage bias $V$ in **b**. The voltage bias $V$ was 17.5 V. **d** Schematic illustration of a graph structure with an intermediate node $N_3$ existing between the nodes $N_1$ and $N_2$. The corresponding edges with different weights are accordingly marked. **e** SEM images of a serial structure composed of two basic modulation units mapping with the graph structure in **d**: global configuration and voltage bias scheme (left panel, scale bar: 100 μm); initial morphology of each unit (middle panel, scale bar: 100 nm); connectivity pattern of conductive filament after memristive switching (right panel, scale bar: 100 nm). **f** Corresponding time-dependent current measurement from the $N_2$ electrode under the voltage bias $V$ in **e**. The voltage bias $V$ (15 V) was applied between the $N_1$ electrode and the $N_2$ electrode while the intermediate node $N_3$ was floating.

agent evolutionary network[45]. Although it may result in certain spatial and temporal variations between the connectivity patterns under the same input conditions, the optimization principle is retained and the results obtained constitute a set of optimal solutions (Supplementary Figs. 15 and 16, Supporting Information). Such stochasticity plays an important role in biological evolution, such as the increase of species abundance caused by gene mutation[46], and hence the coexisting controllability and stochasticity of the MAEN system are desirable.

The parallel and serial structures naturally increase the scale of the MAEN, which can be applied to solution of graph problems. Since the electrodes representing intermediate nodes are floating during operations, they can be replaced by miniaturized metal islands so as to further reduce the device size (see Experimental Section and Supplementary Fig. 17 for fabrication process), where the sizes, locations, and arrangements of the metal islands can affect graph representation. To map the graph structure shown in Fig. 3a, an inert intermediate metal

island $i_1$ ($N_3$) was placed between terminals $T_1$ ($N_1$) and $T_2$ ($N_2$), as shown in Fig. 3b. Since $i_1$ ($N_3$) reduces the equivalent gap distance between $T_1$ ($N_1$) and $T_2$ ($N_2$), that is $d_1 + d_2 < d_3$, the conductive filament prefers to form by way of $i_1$ rather than directly connecting $T_1$ and $T_2$. Indeed, this is verified by the formation of a connectivity pattern in Fig. 3c, d, corresponding to the selection of edges with the smallest weight in total, i.e., $N_1 \overset{2}{\leftrightarrow} N_3 \overset{2}{\leftrightarrow} N_2$ (Fig. 3a). This can be extended to a more complex graph structure in Fig. 3e containing three intermediate nodes ($N_3$, $N_4$, and $N_5$). Figure 3f shows the initial morphology of the corresponding MAEN before electrical stimulation. When the voltage bias $V$ was applied between $T_1$ ($N_1$) and $T_2$ ($N_2$), Ag clusters aligning along $i_2$ ($N_4$) and $i_3$ ($N_5$) won the competition, resulting in filament formation between the $T_1$ ($N_1$) and $T_2$ ($N_2$) by way of $i_2$ ($N_4$) and $i_3$ ($N_5$) (Fig. 3g, h), which is once again consistent with its total smallest weight in the graph, i.e., $N_1 \overset{1}{\leftrightarrow} N_4 \overset{2}{\leftrightarrow} N_5 \overset{1}{\leftrightarrow} N_2$ (Fig. 3e). The modulation effect of metal islands was also supported by kinetic Monte Carlo simulations

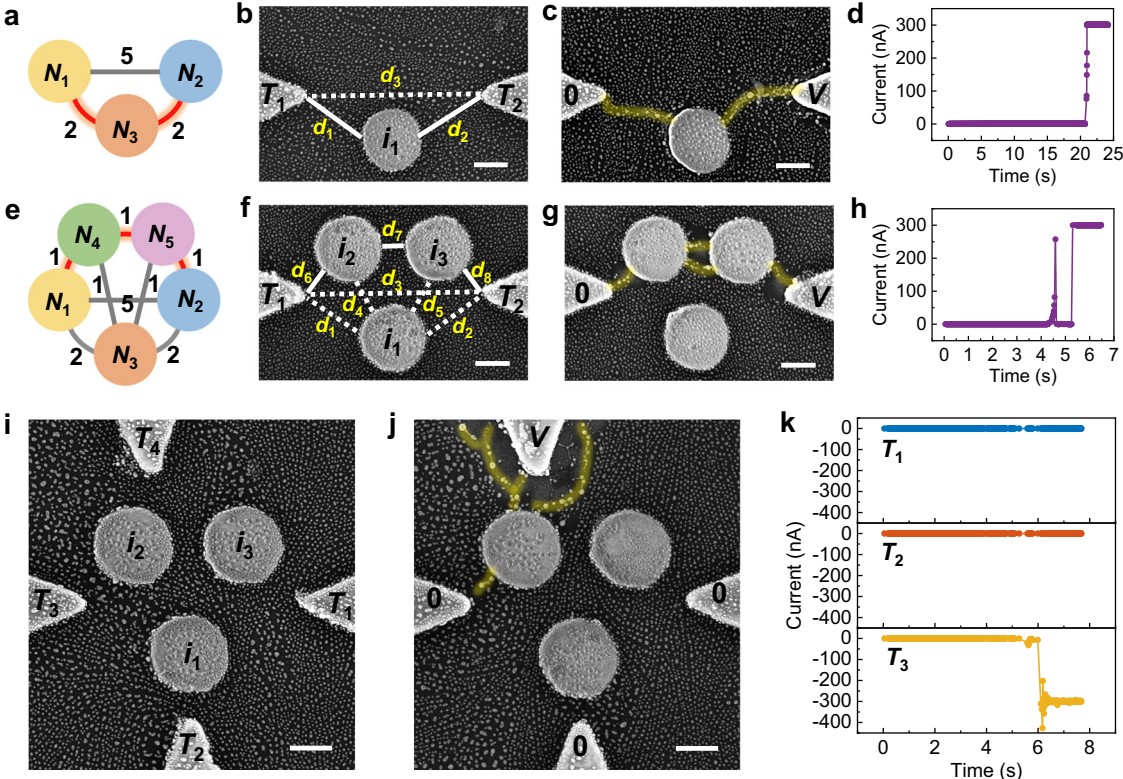

**Fig. 3 | Application of integrated MAENs in the solution of graph problems.**
**a** Schematic illustration of a graph structure with one intermediate node $N_3$ between the nodes $N_1$ and $N_2$. **b** SEM image of the device mapping with the graph structure in **a**. The terminals $T_1$, $T_2$, and metal island $i_1$ are used to represent nodes $N_1$, $N_2$, and $N_3$, respectively. The gap distance mapped with the weight of corresponding edges ($d_1 \leftrightarrow 2$, $d_2 \leftrightarrow 2$, $d_3 \leftrightarrow 5$). Scale bar: 200 nm. **c** SEM image of the device in **b** after memristive switching. Ag clusters self-organize into a conductive filament connecting the terminals $T_1$ and $T_2$ by way of the metal island $i_1$ under the applied voltage bias $V$ (25 V). Scale bar: 200 nm. **d** $I$–$t$ curve corresponding to the experiment in **c**. **e** Schematic illustration of a graph structure with three intermediate nodes $N_3$, $N_4$, and $N_5$ between the nodes $N_1$ and $N_2$. **f** SEM image of the device mapping with the graph structure in **e**. The terminals $T_1$, $T_2$ are used to represent the nodes $N_1$, $N_2$, and the metal islands $i_1$, $i_2$, $i_3$ are used to represent the intermediate nodes $N_3$, $N_4$ and $N_5$, respectively. The gap distance mapped with the weight of corresponding edges ($d_1 \leftrightarrow 2$, $d_2 \leftrightarrow 2$, $d_3 \leftrightarrow 5$, $d_{4\text{-}8} \leftrightarrow 1$). Scale bar: 200 nm. **g** SEM image of the device in **f** after memristive switching. The conductive filament is formed between the terminals $T_1$ and $T_2$ by way of the metal islands $i_2$ and $i_3$ under the applied voltage bias $V$ (20 V). Scale bar: 200 nm. **h** $I$–$t$ curve corresponding to the experiment in **g**. **i** SEM image of the device with four terminals ($T_1$, $T_2$, $T_3$, and $T_4$) and three metal islands ($i_1$, $i_2$, and $i_3$). Scale bar: 200 nm. **j** SEM image of the device in **i** after electrical stimulation. The conductive filament establishes the connection between $T_3$ and $T_4$ by way of $i_2$. The applied voltage $V$ was 15 V. Scale bar: 200 nm. **k** $I$–$t$ curves of terminals $T_1$ (blue color), $T_2$ (orange color), and $T_3$ (yellow color) correspond to the experiment in **j**.

(Supplementary Fig. 18) and verified by consistent experimental results (Supplementary Fig. 19). Supplementary Fig. 20 shows that the same graph problems can be resolved with opposite voltage bias given the switching nature independent on bias polarity. In addition to metal islands, electrode terminals, such as terminal positions (Supplementary Fig. 21), can also be used to improve the flexibility of problem mapping. The parallel and serial structures can also be extended to multiterminal integrated MAEN for the solution of more intricate problems. Figure 3i shows an exemplary graph with four terminals and three intermediate nodes, where the driving voltage $V$ was applied to $T_4$ and the other electrodes ($T_1$, $T_2$, $T_3$) were grounded. The Ag nanoclusters align into a filament by way of $i_2$ and once again find the shortest path between $T_3$ and $T_4$ (Fig. 3j, k). It is worthwhile noting that reusable and reconfigurable computing units are desirable, and thus the computational costs can be effectively saved. As a proof of concept, our experiments have demonstrated the reusability (Supplementary Fig. 22) and reconfigurability (Supplementary Figs. 23 and 24) of the MAEN system for the solution of problems, and the detailed discussions can be seen in the Supplementary Information. Here, the tip of electrode terminals is always sharp to concentrate the electric field, while the metal islands are always circular to achieve uniform regulation on the surrounding electric field in all directions. In order to demonstrate the potential of system generality, the sharp tip of electrode terminals can be replaced by a more rounded shape to share a

similar geometry with the circular metal islands, and the same connectivity patterns can be achieved independent of the electrode geometry, indicating more generalized application (see more detailed discussions in Supplementary Fig. 25, Supporting Information).

## Physical gradient descent in artificial potential field

In fact, the above optimization process based on self-organized evolution of Ag nanoclusters embodies the idea of gradient descent, which is equivalent to constructing a potential distribution function $P(x, y)$ in the corresponding space when the voltage biases are applied to the terminals. In this case, Ag nanoclusters are expected to move toward the direction where the potential drops the fastest and gradually approach the destination terminal with the lowest potential, where the optimization path is indicated by the connectivity pattern formed. Compared with traditional gradient descent algorithms with a single agent performing iterative optimization in the solution space, the present MAEN allows distributed nanocluster agents to physically explore the space with high parallelism, therefore dramatically improving the computing efficiency. In addition, the evolution of Ag nanoclusters in turn changes the potential distribution, i.e., $P(x, y, 0) \rightarrow P(x, y, t)$, forming a feedback mechanism that affects subsequent Ag cluster dynamics. Such feedback provides inherent driving force for persistent evolution of Ag nanoclusters before the emergence of stable connectivity patterns and also accelerates the overall optimization process.

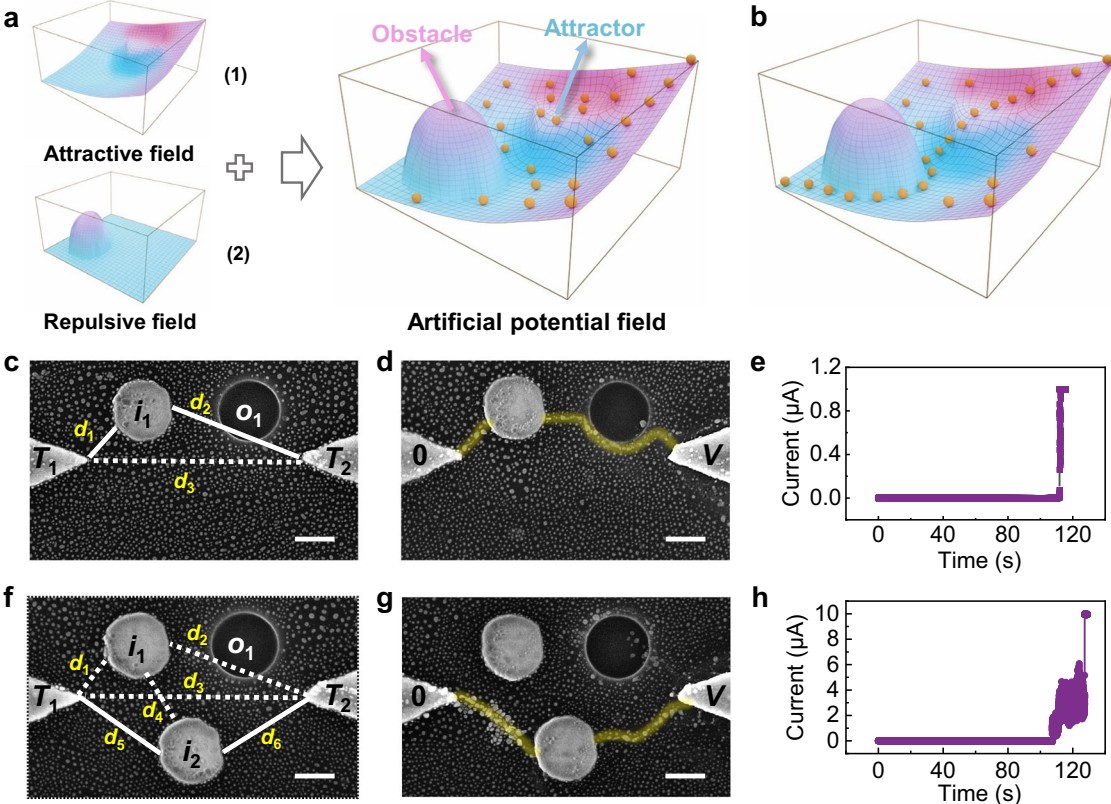

**Fig. 4 | Artificial potential field with superimposed attractive and repulsive components. a** Schematic illustration of an artificial potential field superimposed by the (1) attractive field and (2) repulsive field. **b** Schematic illustration of the final optimal route composed of Ag nanoclusters based on the gradient descent. **c** SEM image of the device with an attractor $i_1$ and an obstacle $o_1$ between the terminals $T_1$ and $T_2$. The shortest path is accordingly marked with a solid line. Scale bar: 200 nm. **d** SEM image of the connection path in (**c**) under the voltage bias $V$. Ag clusters adaptively avoid the obstacle $o_1$ to align into a connective filament along the original shortest path. The applied voltage bias $V$ was 40 V. Scale bar: 200 nm. **e** Time-dependent current measurement corresponding to the experiment in (**d**). **f** SEM image of the device with two attractors $i_1$, $i_2$ and an obstacle $o_1$ between the terminals $T_1$ and $T_2$. The shortest path after considering the obstacle $o_1$ is accordingly marked with the solid line. Scale bar: 200 nm. **g** SEM image of the connection path in (**f**) under the voltage bias $V$. The filament establishes the connection between terminals along the shortest path marked with the solid line in (**f**). The applied voltage bias $V$ was 35 V. Scale bar: 200 nm. **h** Time-dependent current measurement corresponding to the experiment in (**g**).

To testify this idea, we have employed MAEN to construct the artificial field based on gradient descent which is frequently used in classic robot path planning[47]. The complexity of the problem arises from the fact that the artificial potential field usually contains both attractive and repulsive components. The targets and obstacles can be regarded as objects that have attractive and repulsive forces to the agents, and the goal is to guide the agents to avoid obstacles in the potential field with superimposed attractive and repulsive fields and move to the target corresponding to the lowest potential. To implement this in the MAEN, the voltage bias was firstly applied to construct the attractive field, and metal islands are incorporated to effectively modulate the attractive field besides serving as the intermediate nodes. Since the electric field intensity inside the metal islands is approximately zero, it is equivalent to enhancing the electric field intensity around the intermediate nodes, and hence the metal islands can be regarded as immobile attractors for the $Ag^+$ ions, therefore imposing additional attractive field components inside the MAEN. To incorporate repulsive components, ion mobility provides an additional dimension as another considerable kinetic factor regulating the dynamics of Ag nanoclusters in addition to the electric field. As a proof of concept, repulsive centers can be implemented by etching and removing the PEO electrolyte and hence exposing the $SiO_2$ underneath (see Experimental Section and Supplementary Fig. 26). The $Ag^+$ ion mobility of $SiO_2$ is lower than that of self-assembled PEO, but with similar relative permittivity ($\varepsilon_r$)[48], hence the etched region can be regarded as repulsive centers to represent the obstacles. Given the

joint regulations of electric field ($E$) and ion mobility ($\mu$), an artificial potential field with varied ion velocity ($v = \mu E$) can be formed in the MAEN space, as schematically illustrated in Fig. 4a, where the Ag nanoclusters are expected to move towards the direction with maximum ion velocity following the gradient descent principle (Fig. 4b).

Figure 4c shows an artificial potential field with an obstacle $o_1$ existing in the path by way of attractor $i_1$ ($T_1 \overset{d_1}{\leftrightarrow} i_1 \overset{d_2}{\leftrightarrow} T_2$), forming a superimposed space of ion velocity. Experimental results in Fig. 4d, e clearly demonstrate that the Ag nanoclusters spontaneously align into a conductive filament connecting the $T_1$ and $i_1$, followed by extension along the shortest path toward $T_2$. Notably, the Ag nanoclusters adaptively adjust the route and avoid the obstacle $o_1$ in the connectivity pattern. The experimental results are consistent with kinetic Monte Carlo simulations and once again demonstrates the successful solution of the optimization problem using the constructed artificial potential field (Supplementary Fig. 27). Based on the device structure in Fig. 4c, an additional attractor $i_2$ was incorporated (Fig. 4f), so that there exist two shortest paths ($T_1 \overset{d_1}{\leftrightarrow} i_1 \overset{d_2}{\leftrightarrow} T_2$, $T_1 \overset{d_5}{\leftrightarrow} i_2 \overset{d_6}{\leftrightarrow} T_2$) while an obstacle $o_1$ only existed on the upper path ($T_1 \overset{d_1}{\leftrightarrow} i_1 \overset{d_2}{\leftrightarrow} T_2$). When the voltage bias was applied to the MAEN, the Ag nanoclusters were inclined to align through the path without obstacles by way of $i_2$ (i.e., $T_1 \overset{d_5}{\leftrightarrow} i_2 \overset{d_6}{\leftrightarrow} T_2$), as shown in Fig. 4g, h. Since the electric field and ion mobility jointly determine the distribution of artificial potential field, the positions of attractors and obstacles will affect the final connectivity pattern of conductive filaments in general by regulating the artificial potential field with varied ion velocity ($v = \mu E$). However, the

solution result will not be affected in the case of symmetric potential field distribution, as shown in Supplementary Fig. 28. Experimental results shown in Supplementary Fig. 29 have verified the highly reliable property of the MAEN in the physical gradient descent, where the Ag nanoclusters as the agents cooperate and compete with each other based on the positive feedback. This once again demonstrates the adaptability of the MAEN and the successful solution of the optimization problem in the proposed artificial potential field.

It is worthwhile noting that the solution result represented by the connectivity pattern of the filament was mainly obtained by SEM observations, and the time-dependent current measurement was used as an auxiliary readout method, since the detailed connectivity pattern may not be completely reflected through the limited number of probes in the testing probe stations. The reusability and reconfigurability of the MAEN system, as discussed above, can effectively save the overhead caused by the device fabrication for solving different problems, and a dedicated circuit platform including reading and writing periphery (Supplementary Fig. 30) can be developed to probe the MAEN result. In this case, both the gap distance and compliance current can be further reduced, which will also contribute to the reduction of the solution time and therefore further enhancing the computational efficiency. The optimizations in gap distance, compliance current and ion transport properties etc. are expected to be capable of dramatically reducing the power consumption of the MAEN devices. Together with the physics-empowered parallel computing nature of the MAEN system, it implies high potential for enhanced energy efficiency (see more detailed discussion in Supplementary Note 1).

## Discussion

In conclusion, we have experimentally constructed and demonstrated a multi-agent evolutionary network in the form of nanoscale, solid-state, multiterminal device based on the self-organized evolution of distributed Ag nanoclusters under an electric field. The emergence behavior and connectivity of the nanoclusters as distributed agents can be adaptively modulated and effectively controlled for solving graph and optimization problems with high parallelism. Basic modulation units, including distance modulation and voltage modulation, were demonstrated, along with a hybrid mode for joint modulation. In addition, both parallel and serial integrations of the basic units are experimentally demonstrated, implying scaling potential of the MAEN toward large-scale systems. The integrated MAEN has been successfully applied to the solution of two- and multiterminal graph problems. An artificial potential field superimposed by both attractive and repulsive components with varied ion velocity is constructed, showing self-adaptive route planning ability with obstacle avoidance based on gradient descent of the MAEN structure. The devices in this work were constructed to demonstrate the principles of MAEN system based on the self-organized evolution of Ag nanoclusters. In order to meet the demands of practical applications, the reusability, reconfigurability and energy efficiency of the MAEN system should be further improved. Taking full advantage of the coexisting controllability and stochasticity in swarm intelligence, the massively parallel, self-adaptive and high-throughput information processing in our MAEN system provides an encouraging pathway toward energy-efficient computing hardware.

## Methods

### Device fabrication

The devices were fabricated on the silicon substrate with 300 nm thick thermally grown silicon dioxide. The substrates were initially cleaned using acetone and isopropanol successively for 15 min ultrasonication. Then, Ti (1 nm) and Au (40 nm) as the electrodes or metal islands were patterned on the substrate by electron beam lithography using PMMA as the resist, followed by electron beam evaporation and lift-off. To get the polymer thin film, Polyethylene oxide (PEO) with a molecular weight of 100,000 g mol$^{-1}$, was initially dissolved in acetonitrile solvent by 5wt‰ for 1 h ultrasound at room temperature to acquire homogeneous solution. Then, the solution was dropped and spin-coated on the surface of devices. The solvent evaporated during the high-speed spin-coating process, and finally the solid polymer thin film (~25 nm) was formed. Finally, discrete Ag nanoclusters were partly incorporated into the PEO by electron beam evaporation with a thickness of 3 nm below the filming condition. The polymer electrolyte material PEO was etched to expose the $SiO_2$ layer area as the obstacle generating repulsive field by using dual-beam focused ion beam system (FIB Helios G4 UX).

### Device characterization

All the electrical measurements were carried out at atmospheric pressure and ambient temperature using an Agilent B1500A semiconductor parameter analyzer in a probe station. The constant voltage biases were applied to the terminals to monitor time-dependent current curves, while the DC $I–V$ sweep was carried out by applying the uniformly varying voltage bias. The SEM imaging of the devices was performed before and after network evolutions by Helios NanoLab 600i at 5 KV, while the topography of the Ag nanoclusters and filaments was characterized by the PeakForce tapping mode of AFM (Bruker Dimension Icon). The AFM data were analyzed by the NanoScope Analysis 1.4 software (Bruker).

### Simulation

The simulation was performed using the Matlab software with a kinetic Monte Carlo (MC) method. During the simulation, three physical processes were considered: (1) Ag ionization from the clusters, (2) ionic transport in the electrolyte, and (3) the reduction of $Ag^+$ ions. The current and electric field distribution were obtained based on a resistance network model by solving Kirchhoff's equations. The evolution of metal atoms distribution, current and electric field with time were iteratively simulated and updated. More details and parameters about the simulation can be found in Supplementary Fig. 5 (Supporting Information).

## Data availability

All data supporting this study and its findings are available within the article, its Supplementary Information and associated files. The source data underlying Figs. 1f, k, n, r–t, v, 2c, f, 3d, h, k, 4e, h have been deposited at https://zenodo.org/record/6641560#.YqhPdNpBzb0 or are available from the corresponding author upon reasonable request.

## Code availability

The codes used for the simulations are described in https://github.com/liyingxu001/MC-simulation-codes or are available from the corresponding author upon reasonable request.

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

## Acknowledgements

This work was supported by the National Key R&D Program of China (2017YFA0207600), National Natural Science Foundation of China (61925401, 92064004, 61927901, 92164302), Project 2019BD002 and 2020BD010 supported by PKU-Baidu Fund, and the 111 Project (B18001). Y.Y. acknowledges the support from the Fok Ying-Tong Education Foundation and the Tencent Foundation through the XPLORER PRIZE.

## Author contributions

Y.Y. conceived the project. L.X. and J.Z. fabricated the devices and performed all experiments. B.C. performed the Monte Carlo simulations. Z.Y., K.L., B.D., and T.Z. contributed to the analysis of the data. L.X. and Y.Y. prepared the manuscript. Y.Y. and R.H. directed all the research and supervised this work. All authors analyzed the results and implications and commented on the manuscript at all stages.

## Competing interests

The authors declare no competing interests.
