## [Peer Review File · Nature Communications]

REVIEWER COMMENTS

Reviewer #1 (Remarks to the Author):

The manuscript by Xu et al. reports a hardware implementation of the multi-agent system using dynamics of Ag nanoclusters., which demonstrates gradient descent route planning and self-adaptive obstacle avoidance. The results are interesting and timely, given the physics-based neuromorphic computing is very topical nowadays. The manuscript is systematically organized with well-prepared plots. A few technical comments are as following.

1. In Fig. 1u, the author mentioned 'The gap distance from left to right is 100 nm, 200 nm, 200 nm, 300 nm, 400 nm, 500 nm, 600 nm and 700 nm, respectively.' However, the SEM images seem to reveal that the gap distance of T3 is smaller than T2 while the gap distance between T4 and T5 is close.

2. It seems the compliance current may also be an importance parameter to play with (Fig. S6). Could the authors suggest how to determine the proper compliance current for different physical structures (Fig. 2-4)?

3. What could be the advantages and disadvantages if voltage modulation scheme is employed in Fig. 2-4?

4. For the last experiment, as shown in Fig. 4f-h, will the position of attractor i_1 and obstacle o_1 affect the results? If their positions are swapped, will that yield the same result?

5. Could the authors comment on whether the same graph problems shown in Fig. 3 and Fig. 4 can be resolved with opposite voltage bias given the bipolar nature of switching?

6. What's the minimum gap between metal nodes? Will a smaller gap benefit the stability since Ag nanocluster may also feature interface energy driven surface diffusion, such as that shown in the diffusive memristors?

Reviewer #2 (Remarks to the Author):

In this manuscript, Xu et al. propose a nanoscale, solid-state multi-agent evolutionary network composed of memristive structures based on Ag nanoclusters. This hardware system can be used to explore the solution spaces of optimization problems with high parallelism via the electrochemical dynamics of Ag nanoclusters under electric field. And the authors experimentally demonstrate that this type of hardware system can successfully solve some simple graph problems and route optimizations of artificial potential field. The manuscript is well structured. The experiments are clearly demonstrated. The intrinsic parallel computing capability of the hardware system is very attractive. The authors have shown beautiful results experimentally with great scientific interests. I would recommend for publication after addressing the following issues:

Issues

- Within the experimental demonstration, the hardware system seems to be not reusable or reconfigurable. And it would be very expensive and time-consuming to build one system for each problem we want to solve. Thus, can the authors comment on if it is feasible to reuse or reconfigure the system for realistic problems? For instance, There are previous work on diffusive memristors using similar material systems and showing the Ag ion tends to go back to their original positions under zero electrical bias, driven by the interfacial energy minimization between Ag and dielectrics. This volatile behavior can recover the system to its initial state for being reused for new problems, which may be worth a discussion. According to Figure S7, the memristors are volatile and can be repeatedly switched. However, all the systems in their manuscript are switched only once for problem solving and it was not shown if the systems can be reused after that or the ones used in the manuscript were nonvolatile version? In addition, how about the reconfigurability for solving a very different problem?
- Though the time complexity of Ag nanocluster reorganization is $O(1)$, the time that is needed to fabricate the structures and to visualize the optimal path is tremendous compared to conventional computers. And the advantage of this hardware system would be significantly reduced in this case. Thus, can the authors discuss the potential of solving this problem?
- In Figure 3, the electrodes with voltage applied on always have sharp tips while the metal island in the middle are always circular pads. This seems to reduce the generality of the hardware system. If all the electrode pads of a hardware system share the same shape, then the system will be able to compute the optimal route between each two points, and the capability of the whole system will be much more promising.
- It is shown in Figure 1 that the basic modulation units can be controlled by both voltage amplitude and electrode distance. However, only electrode distance modulations are used within the following solution of graph problems and artificial potential field. Can the authors show how voltage amplitude modulation can play a role during solving the problems?

Reviewer #3 (Remarks to the Author):

The paper deals with a network based on self-organizing Ag nanoclusters and the use of induced electrical paths as optimized solutions of graph problems and gradient descent. It is well written, interesting and complete, also considering the huge supplementary material. Furthermore, it is timely, since the field of designless computing with similar networks based on nanowires, nanoparticles or random dopants is becoming much popular every day.

I have two main requests that I think should be satisfied before proceeding with the publication:

- Although being an interesting and complete paper, no comparison at all is done with current literature employing similar networks based on nanowires, nanoparticles or random dopants. I think a deep discussion should be done versus works of Brown, van der Wiel and Gimzewski (at least).

- Even if it is claimed just as an “encouraging pathway toward energy efficient computing hardware”, a discussion on power consumption is needed. Honestly, it is convincing that optimal paths are created, but considering applied voltage (tens of V), timescale (tens of seconds) and measured current (hundreds of nA to several microamps), the power consumption looks huge (from micro to even milli Joule?). How could it be reduced, to follow the above-mentioned claim?

Other minor points are:

- Fig S1 is never cited in the main text

- Fig S3 is claimed to show the path from AFM, but honestly I don't see it: what should I see in the pictures?

- Memristive behavior is just shown in some I-V plots in Fig S7. I think some data about endurance and retention would be needed, since from that picture the I-V plots look not that repeatable.

- Is there any (short) memory effect? If yes, can it be used for computing?

- How the simultaneously writing/reading would be guaranteed in the perspective of several more inputs/outputs?

- It is not clear to me why both V amplitude and distance are used: since driving force is the applied electric field, there's no difference in principle in using one or the other, but tuning V is far easier than changing distance by lithography. Please clarify. The metaphor of cost (distance) and reward (voltage) looks obscure to me, since here the two variables are not really independent in forming E.

MS No: NCOMMS-22-03875-T

Title: A Distributed Nanocluster Based Multi-Agent Evolutionary Network

Response to the editor and the reviewers

We would like to sincerely thank the editor for the kind consideration of our manuscript and the reviewers for their constructive suggestions, which are very valuable in improving our manuscript. We have carefully considered all the reviewers' questions and made corresponding revisions. We hope the editor and reviewers will find the revised manuscript suitable for publication in *Nature communications*. The point-to-point responses and changes made are listed below.

Reviewer #1 (Remarks to the Author):

Overall Remarks: The manuscript by Xu et al. reports a hardware implementation of the multi-agent system using dynamics of Ag nanoclusters, which demonstrates gradient descent route planning and self-adaptive obstacle avoidance. The results are interesting and timely, given the physics-based neuromorphic computing is very topical nowadays. The manuscript is systematically organized with well-prepared plots. A few technical comments are as following.

Our response: We would like to sincerely thank the reviewer for the positive evaluation and valuable suggestions. We have carefully addressed all the points, as shown below.

1. In Fig. 1u, the author mentioned 'The gap distance from left to right is 100 nm, 200 nm, 200 nm, 300 nm, 400 nm, 500 nm, 600 nm and 700 nm, respectively.' However, the SEM images seem to reveal that the gap distance of T_3 is smaller than T_2 while the gap distance between T_4 and T_5 is close.

Our response: We would like to thank the reviewer for raising the question. The nine-terminal device shown in **Fig. 1u** under the hybrid distance-voltage modulation mode was used to demonstrate the potential that the number of terminals in the basic modulation unit can be flexibly extended according to actual demands. "*The gap*

distance from left to right is 100 nm, 200 nm, 200 nm, 300 nm, 400 nm, 500 nm, 600 nm and 700 nm, respectively” described in the original text refers to the designed gap distance in the device layout. Due to inevitable variations in device fabrication process, such as that in electron beam lithography (EBL) and lift-off, there may exist a certain deviation of the actual gap distance from the designed gap distance. Here we compared the gap distance of T_2 , T_3 and the gap distance of T_4 , T_5 in the SEM image, as can be seen from the yellow horizontal lines in **Fig. R1**, where the gap distance of T_2 is slightly smaller than that of T_3 while the gap distance of T_4 is slightly smaller than T_5 . The slightly different gap distances between T_2 and T_3 originates from variations in device fabrication process, while the gap distances of T_4 and T_5 are in line with the designed values. As a result, the gap distances of the terminal can be described as “ $T_2 < T_3 < T_4 < T_5$ ”, while the voltage biases applied to the terminal can be describe as “ $T_2 < T_3 = T_4 = T_5$ ” ($V_2 = 25$ V, $V_3 = 15$ V). Although the gap distance of T_2 is slightly smaller than that of T_3 , under the collaborative regulation of gap distance and voltage bias the conductive filament preferentially establishes connections between the terminals T_3 and T_9 , where the evolution of Ag clusters is most obvious driven by the higher electric field compared with other terminals. Such result is consistent with the expected result corresponding to the designed gap distances and applied voltages.

Figure R1 | Comparison of the gap distance between the T_2 and T_3 and the gap distance between the T_4 and T_5 of the nine-terminal device shown in Fig. 1u. The yellow horizontal lines clearly mark the relative positions of the terminals.

In order to address this question, the figure caption of **Fig. 1u** has been corrected to “*The designed gap distance from left to right is 100 nm, 200 nm, 200 nm, 300 nm, 400 nm, 500 nm, 600 nm and 700 nm, respectively*”.

2. It seems the compliance current may also be an importance parameter to play with (Fig. S6). Could the authors suggest how to determine the proper compliance current for different physical structures (Fig. 2-4)?

Our response: We would like to thank the reviewer for raising the question. The compliance current is indeed an important parameter in modulating the self-organized evolution of Ag nanoclusters. From **Fig. S6** we can see that the compliance current will affect the filament morphology, so that it is very important to select appropriate compliance current during the electrical stimulation. **Fig. R2** shows typical compliance current conditions adopted during the testing process in the basic modulation units (data in Fig. 1o,r), parallel structures (data in Fig. 2b) and serial structures (data in Fig. 2e), respectively.

For the basic modulation units (e.g. **Fig. 1d,k,n,o-q,u**) and parallel structures (e.g. **Fig. 2b, Fig. S14, S16**), a compliance current between 10-50 nA is adopted in most cases. For the problems containing serial structures (e.g. **Fig. 2e, Fig. 3c,g,j, Fig. 4d,g**), a compliance current of 0.3-10 μ A is usually needed to provide larger driving force. Although the specific value of compliance current needs to be selected according to the specific problem, it is overall higher than that in basic modulation units and parallel structures due to the generally longer gap distance.

In order to describe in detail how to determine the proper compliance current for different physical structures, we have added the following sentences in **Page 33** of the revised manuscript “*In this work, different values of compliance current are selected for different physical structures. For the basic modulation units and parallel structures, a compliance current between 10-50 nA is adopted in most cases. For the problems containing serial structures, larger compliance current of 0.3-10 μ A is usually needed*”

to provide larger driving force, due to the generally longer gap distance in serial structures.”

Figure R2 | Typical compliance current of basic modulation units and integration structures. **a**, A case of basic modulation unit including the connectivity pattern and corresponding compliance current (left panel of SEM, Scale bar: 50 μm ; middle and right panels of SEM, Scale bar: 200 nm). **b**, A case of parallel structure including the connectivity pattern and corresponding compliance current (left panel of SEM, Scale bar: 50 μm ; middle and right panels of SEM, Scale bar: 100 nm). **c**, A case of serial structure including the connectivity pattern and corresponding compliance current (left panel of SEM, Scale bar: 100 μm ; middle and right panels of SEM, Scale bar: 100 nm).

3. What could be the advantages and disadvantages if voltage modulation scheme is employed in Fig. 2-4?

Our response: We would like to thank the reviewer for raising the question. The distance modulation and the voltage modulation are two basic modulation schemes of the MAEN system, which are used to represent different input information and allows flexible application. In fact, the two basic modulation schemes have their own advantages and disadvantages in terms of mapping and solution of the problems.

For the distance modulation scheme, the graph information is directly mapped with the device structure, and the weights of edges are represented by regulating the positions of electrode terminals or metal islands. The advantages and disadvantages include: 1) When solving the problems, the same voltage biases are applied to the terminals corresponding to the same node, so that these terminals can be connected to a common electrode with constant bias for saving area overhead; 2) Since the weights of edges are directly reflected in the device structure, the problems can be conveniently and efficiently solved in one step by applying voltage biases between the electrodes representing the specified nodes while the intermediate nodes remain floating; 3) The disadvantage is different device structures need to be used to map different graph information, resulting in poor reconfigurability.

In case of the voltage modulation scheme, the weights are represented by the voltage biases applied to the electrode terminals. The advantages and disadvantages include: 1) The same device structure can be used to represent different graph problems, hence leading to higher reconfigurability; 2) Since each weight needs to be represented by the voltage bias applied to the corresponding electrode, the device will occupy a large area when the number of edges is large.

To meet the requirements in different application scenarios, an optimal modulation scheme can be designed to improve the efficiency and flexibility for problem solution by tuning gap distance d and voltage bias V separately or collectively based on the respective characteristics of the two modulation units.

To clarify the advantages and disadvantages of the two basic modulation units, the following sentences have been added into **Page 8** of the revised manuscript “*The distance and voltage modulation units as two fundamental building blocks of MAEN system have their own advantages and disadvantages in terms of mapping and solution of the problems. For the distance modulation unit, the graph information is directly mapped with the device structure, which facilitates efficient solution of problems and saves overhead in electrode area, but leads to poor reconfigurability since the weights are represented in hardware. On the contrary, for the voltage modulation unit, the weights are represented by the voltage biases applied to the electrode terminals, so that*

the same device structure can be used to represent different graph problems, hence leading to higher reconfigurability. To meet the requirements in different application scenarios, an optimal modulation scheme can be designed to improve the efficiency and flexibility for problem solution by tuning gap distance d and voltage bias V separately or collectively based on the respective characteristics of the two modulation units. Under the hybrid distance-voltage modulation scheme, different input information can be directly integrated in the MAEN, thus enhancing the computing efficiency.”

4. For the last experiment, as shown in Fig. 4f-h, will the position of attractor i_1 and obstacle o_1 affect the results? If their positions are swapped, will that yield the same result?

Our response: We would like to thank the reviewer for raising this interesting question. The attractive field of the artificial potential problem is constructed based on the electric field. By applying a voltage bias between the electrode terminals, the potential gradient serves as the attractive force to drive Ag^+ ions toward the target. The metal islands can be regarded as immobile attractors to effectively modulate the attractive field distribution by imposing additional attractive field components inside the MAEN, while as another considerable kinetic factor regulating the dynamics of Ag nanoclusters the ion mobility is used to incorporate repulsive components ($v = \mu E$). Taking PEO region as a reference, the Ag^+ ion velocity is relatively slower in the SiO_2 region since the ion mobility is lower, which can be regarded as the obstacle to form repulsive force hindering Ag^+ ion migration. Therefore, both the positions of attractor and obstacle can directly affect the potential field distribution, and the connectivity pattern of conductive filament following the gradient descent principle will be adjusted along with the change of potential field gradient. However, the result will not be affected if the position of attractor i_1 and obstacle o_1 shown in **Fig. 4f** are swapped, since the electric field distribution will be symmetric, leading to the same cluster evolution result. To verify this, we have performed further Monte Carlo simulation on this case, the results clearly demonstrate that the same connectivity pattern is obtained, as shown in **Fig. S28**.

Figure S28 | The Monte Carlo simulation of the artificial potential field shown in Fig. 4f before and after swapping the position of attractor i_1 and obstacle o_1 . a, The Ag atom/cluster evolution processes in Fig. 4f when the voltage bias was applied between the terminals T_1 and T_2 . The conductive filament was connected along the path $T_1 \xleftrightarrow{d_5} i_2 \xleftrightarrow{d_6} T_2$, which is consistent with the experiment result shown in Fig. 4g. b, The Ag atom/cluster evolution processes after swapping the position of attractor i_1 and obstacle o_1 in Fig. 4f. The conductive filament still establishes connection between the terminals T_1 and T_2 through the path without obstacles by way of i_2 .

In order to address this question, we have added the new data as **Figure S28** and the following sentences in section “**The effect of position exchange of attractor i_1 and obstacle o_1 on the solution result**” (Supplementary Information) in **Page 52** of the revised manuscript: “*Here, we performed Monte Carlo simulation where the position of attractor i_1 and obstacle o_1 are swapped in the artificial potential field problem shown in Fig. 4f. From Fig. S28 one can see that the solution result is not affected since the electric field distribution will be symmetric, leading to the same cluster evolution result. Consistent with the result shown in Fig. 4g, the filament still establishes connection between the terminals T_1 and T_2 through the path without obstacles by way of i_2 (i.e. $T_1 \xleftrightarrow{d_5} i_2 \xleftrightarrow{d_6} T_2$).*”

In addition, we added the following discussions in **Page 13** of the revised manuscript “*Since the electric field and ion mobility jointly determine the distribution*

*of artificial potential field, the positions of attractors and obstacles will affect the final connectivity pattern of conductive filaments in general by regulating artificial potential field with varied ion velocity ($v = \mu E$). However, the solution result will not be affected in case of symmetric potential field distribution, as shown in **Supplementary Fig. 28**.”*

5. Could the authors comment on whether the same graph problems shown in Fig. 3 and Fig. 4 can be resolved with opposite voltage bias given the bipolar nature of switching?

Our response: We would like to thank the reviewer for raising the interesting question. The reviewer is absolutely correct that the same problems can be resolved with opposite voltage bias given the bipolar nature of switching. In order to prove this, we have interchanged the voltage bias applied to terminals T_1 and T_2 in **Fig. 3g**. The result shown in **Fig. S20** demonstrates that Ag nanoclusters physically evolving along the metal islands i_2 (N_4) and i_3 (N_5) still won the competition, and the conductive filament established connection between the terminals T_1 (N_1) and T_2 (N_2) by way of the islands i_2 (N_4) and i_3 (N_5).

To clarify this point, we have added **Figure S20** in **Page 45** and included the following sentences in **Page 11** of the revised manuscript “*The modulation effect of metal islands was also supported by kinetic Monte Carlo simulations (Supplementary Fig. 18) and verified by consistent experimental results (Supplementary Fig. 19). Supplementary Fig. 20 shows that the same graph problems can be resolved with opposite voltage bias given the switching nature independent on bias polarity.*”

Figure S20 | Solution results with opposite voltage bias in Fig. 3g (i.e. the voltage bias V is applied to the terminal T_1 while the terminal T_2 was grounded). The

definition of terminals is shown in **Fig. 3f**. **a**, SEM image of the filament morphology after memristive switching. Ag nanoclusters physically evolving along the metal islands i_2 (N_4) and i_3 (N_5) still won the competition, and the conductive filament established connection between the terminals T_1 (N_1) and T_2 (N_2) by way of the metal islands i_2 (N_4) and i_3 (N_5) under the applied voltage bias V (35 V). Scale bar: 200 nm. **b**, I - t curve corresponding to the experiment in **(a)**.

6. What's the minimum gap between metal nodes? Will a smaller gap benefit the stability since Ag nanocluster may also feature interface energy driven surface diffusion, such as that shown in the diffusive memristors?

Our response: We would like to thank the reviewer for raising the important question. Indeed, the interfacial energy driven surface diffusion of Ag nanoclusters, as shown in the diffusive memristors proposed by Z. Wang et al (Ref. R1), also exists in our device. This can be shown through the I - V measurement with threshold switching in **Fig. S7**, where the device spontaneously relaxed back to the off state after removing the voltage bias, which is exactly due to the filament dissolution facilitated by interfacial energy minimization with diffusion mechanism. The minimum gap between metal nodes in our experimental demonstration is ~ 100 nm, which is for the sake of clear SEM observation so as to demonstrate the correct solution from MAEN in this work. Future electrical probing periphery can be incorporated and used to extract the final result, as depicted in **Fig. S30** of the revised manuscript. In this case, it is unnecessary to use relatively large gap distance for observation, so smaller gaps can be utilized to improve the structure stability of filaments. A smaller gap is also beneficial for the decrease of threshold voltage and solution time, thus saving the computational costs.

In order to clarify this point, we have included the following discussion in **Page 6** of the revised manuscript “*The threshold switching characteristics indicate the device spontaneously relaxed back to the off state after removing the voltage bias, which is exactly due to the filament dissolution facilitated by interfacial energy minimization between Ag nanoclusters and dielectrics^{42,43}.*”

and in **Page 35** “*The incomplete rupture of conductive filament can reflect the*

memory effect of device to some extent, which can be used to read out the solution result by the structural observation as well as electrical measurement. The structure stability of conductive filament can be further tuned by decreasing the gap distance to limit interfacial energy driven surface diffusion. In addition, a smaller gap will be beneficial for the decrease of threshold voltage and solution time, thus saving the computational costs.”

Reviewer #2 (Remarks to the Author):

Overall Remarks: In this manuscript, Xu et al. propose a nanoscale, solid-state multi-agent evolutionary network composed of memristive structures based on Ag nanoclusters. This hardware system can be used to explore the solution spaces of optimization problems with high parallelism via the electrochemical dynamics of Ag nanoclusters under electric field. And the authors experimentally demonstrate that this type of hardware system can successfully solve some simple graph problems and route optimizations of artificial potential field. The manuscript is well structured. The experiments are clearly demonstrated. The intrinsic parallel computing capability of the hardware system is very attractive. The authors have shown beautiful results experimentally with great scientific interests. I would recommend for publication after addressing the following issues:

Our response: We would like to sincerely thank the reviewer for the positive evaluation and the valuable suggestions. We have carefully addressed all the points, as shown below.

Issues

1. Within the experimental demonstration, the hardware system seems to be not reusable or reconfigurable. And it would be very expensive and time-consuming to build one system for each problem we want to solve. Thus, can the authors comment on if it is feasible to reuse or reconfigure the system for realistic problems? For instance, there are previous work on diffusive memristors using similar

material systems and showing the Ag ion tends to go back to their original positions under zero electrical bias, driven by the interfacial energy minimization between Ag and dielectrics. This volatile behavior can recover the system to its initial state for being reused for new problems, which may be worth a discussion. According to Figure S7, the memristors are volatile and can be repeatedly switched. However, all the systems in their manuscript are switched only once for problem solving and it was not shown if the systems can be reused after that or the ones used in the manuscript were nonvolatile version? In addition, how about the reconfigurability for solving a very different problem?

Our response: We would like to thank the reviewer for the extremely valuable suggestion. In the following discussion, we will address the reusability and reconfigurability of the MAEN system separately.

Reusability

As the reviewer kindly pointed out, the Au/SiO_xN_y:Ag/Au diffusive memristors reported in prior works by Z. Wang et al (Ref. R1) exhibit relaxation effects, where the conductive filament will quickly shrink to spherical Ag nanoclusters driven by interfacial energy minimization when the external biasing is removed. Such volatile behavior of diffusive memristors may recover the system and make it reusable. The reviewer is also absolutely correct that our devices are volatile and show threshold switching I - V loops (**Fig. S7**), which is due to the spontaneous diffusion dynamics driven by the interfacial energy minimization between Ag nanoclusters and dielectrics. The electrical volatility indicates that the conductive filament has spontaneously broken into discrete Ag clusters. This therefore provides substrate for reusing the same device for the solution of problems.

To demonstrate this, we have performed further experiment to verify the reusability of the MAEN system, as can be seen in **Fig. S22**, where it shows a consecutive sequence of operations on the same MAEN system with 4 terminals and 3 metal islands. For each step, the conductive filament can establish connection along the optimal path between the corresponding terminals, and the spontaneous relaxation of particles after removing the biases allows the device to be reused and operate correctly

in the next steps, as shown in **Fig. S22**, therefore demonstrating reusability of the MAEN devices. In the future, material systems with large wetting contact angles, such as $\text{MgO}_x\text{:Ag}$, $\text{SiO}_x\text{N}_y\text{:Ag}$ and $\text{HfO}_x\text{:Ag}$ (Ref. R1), may be considered. It is reported that the conductive filament may quickly shrink to original Ag nanoclusters driven by interfacial energy in these material systems when the external biasing is removed, so that the reusability of system can be further improved.

Figure S22 | Sequential operations on the MAEN systems with 4 terminals (T_1 , T_2 , T_3 and T_4) and 3 metal islands (i_1 , i_2 and i_3). **a**, SEM image of the device structure before electrical stimulation. Scale bar: 200 nm. **b**, SEM image of the filament morphology when the voltage bias V_1 (25 V) was applied between the terminals T_1 and T_3 . The conductive filament was formed between the terminals T_1 and T_3 by way of the metal islands i_2 and i_3 under the applied voltage bias V_1 . Scale bar: 200 nm. **c**, SEM image of the filament morphology when the voltage bias V_1 (25 V) was applied between the terminals T_1 and T_3 , followed by a voltage bias V_2 (10 V) between the terminals T_3 and T_4 . Correspondingly, the conductive filament was connected between the terminals T_3 and T_4 along the metal island i_2 under the voltage bias V_2 in addition to the connection path shown in **(b)** under the voltage bias V_1 . Scale bar: 200 nm. **d**, SEM image of the filament morphology after successively applying three voltage bias schemes (i.e. the voltage bias V_1 (15 V) between T_1 and T_3 , the voltage bias V_2 (5 V) between T_3 and T_4 and the voltage bias V_3 (5 V) between T_2 and T_4). Since there exists two optimal path

between the terminals T_2 and T_4 under the voltage bias V_3 , the conductive filament was both formed by way of the metal islands i_1 and i_2 and by way of the metal islands i_1 and i_3 in addition to the connection path shown in (c) under the voltage bias V_1 and V_2 . Scale bar: 200 nm. The subscripts of applied voltage bias labelled in the figures represent the order of three voltage bias schemes.

Reconfigurability

The MAEN system possesses potential for high reconfigurability. **Fig. S23** gives an example to show that different problems can be solved with the same device structure. We can see that under two different voltage bias schemes, Ag nanoclusters spontaneously align into a conducting filament along the optimal path, leading to two different connectivity patterns. In the case of **Fig. S23a**, V and 0 were applied to the neighboring terminals T_1 and T_4 respectively, while the other two terminals (i.e. T_2 and T_3) were biased at $1/2V$. SEM observation and $I-t$ curves suggest that an electrical connection was formed between the terminals T_1 and T_4 by way of the metal island i_3 . In another case of **Fig. S23b**, V and 0 were applied to the diagonal terminals T_1 and T_3 , respectively, while the other two terminals (i.e. T_2 and T_4) were assigned to $1/2V$. Compared with the previous case, a different pattern connecting the terminals T_1 and T_3 by way of the metal islands i_2 and i_3 was formed. In addition, the graph problem shown in **Fig. 3e** can also be solved by using the device structure shown in **Fig. S23**, where the original two-terminal device structure (**Fig. 3f**) is included as a sub-graph.

Figure S23 | Connectivity patterns under two different voltage bias schemes in the same device structure with 4 terminals (T_1 , T_2 , T_3 and T_4) and 3 metal islands (i_1 , i_2 and i_3). **a**, One case where V and 0 were respectively applied to the neighboring terminals T_1 and T_4 , while the other two terminals (i.e. T_2 and T_3) were biased at $1/2V$: SEM images of device morphology before memristive switching (upper left panel, scale bar: 200 nm) and after applying the voltage bias scheme (upper middle panel, scale bar: 200 nm); corresponding time-dependent current measurement from 4 terminals (upper right panel), and Monte Carlo simulation (bottom panels). The applied voltage bias V was 15 V. The conductive filament finally established connection between the terminals T_1 and T_4 by way of metal island i_3 . **b**, Another case where V and 0 were respectively applied to the diagonal terminals T_1 and T_3 , while the other two terminals (i.e. T_2 and T_4) were assigned to $1/2V$: SEM images of device morphology before memristive

switching (upper left panel, scale bar: 200 nm) and after applying the voltage bias scheme (upper middle panel, scale bar: 200 nm); corresponding time-dependent current measurement from 4 terminals (upper right panel), and Monte Carlo simulation (bottom panels). The applied voltage bias V was 30 V. The conductive filament was finally formed between the terminals T_1 and T_3 by way of metal islands i_2 and i_3 .

It is worthwhile noting that distance modulation scheme and voltage modulation scheme as two basic modulation units, are used to build the MAEN system in this work, and may also be used to achieve reconfigurability. For the voltage modulation scheme, the weights are represented by the voltage biases applied to the electrode terminals, so that the same device structure can be used to represent different graph problems, hence leading to higher reconfigurability, compared with the distance modulation scheme. Therefore, the voltage modulation scheme can be used to enable system reconfigurability, where different voltage biases representing weight information can be applied to the same device structure, leading to different connectivity results corresponding to the specific problems.

In future research works, we propose that the back gates can be further added to achieve gate modulation. Once the back gates are incorporated, the weight can be represented by the gate control signal, which provides another degree of freedom to effectively regulate cluster evolution and filament growth. The gate modulation scheme combines the advantages of distance modulation and voltage modulation schemes, simultaneously possessing high efficiency, high flexibility and high reconfigurability for solution of problems. **Fig. S24** shows an example of solving a 3×3 maze problem. A five-terminal gate modulation unit is used to represent the connectivity equivalent to weight information between each grid and adjacent grids in the maze. Each gap distance between the terminals in the unit keeps the same. To map the maze problem shown in **Fig. S24a**, a MAEN device is constructed by cascading the basic gate modulation units using parallel and serial schemes, as shown in **Fig. S24b**. If one wants to find the shortest path between the grid 1 and grid 9, a voltage bias can be applied between the terminals representing grid 1 and grid 9, while the control voltage signals representing

the weight information are applied to the corresponding gates, and thus the time complexity of solving the problem is only $O(1)$. Thanks to the same device structure independent of the specific maze, a system mapping the $N \times N$ maze is capable of solving arbitrary $n \times n$ maze problems ($n \leq N$), suggesting high reconfigurability of the system.

Figure S24 | A 3×3 maze problem solved by the gate modulation scheme. a, The connectivity of maze. The white grid represents connected and the grey grid represents disconnected. **b,** The MAEN system with topological structure capable of mapping the 3×3 maze by cascading the basic gate modulation units using parallel and serial schemes. The enlarged image on the left exemplifies the gate modulation unit representing the connectivity of grid 1 in (a).

In order to address the reusability issue clearly, we have included the new results in Supplementary **Figure S22**, along with the following sentences in **Page 46** of the revised manuscript “*The spontaneous diffusion dynamics of conductive filament driven by interfacial energy minimization between the Ag nanoclusters and dielectrics (Supplementary Ref. S2 and S5) causes electrical disconnection between the electrode terminals after removing the external biasing, which provides substrate for reusing the same MAEN device for the solution of problems. Fig. S22 shows a consecutive sequence of operations on the same MAEN system with 4 terminals and 3 metal islands. For each step, the conductive filament can establish connection along the optimal path between the corresponding terminals, and the spontaneous relaxation of particles after removing the biases allows the device to be reused and operate correctly in the next steps. It is worth mentioning that the previous connectivity patterns may have a certain impact on the subsequent solutions. In the future, material systems*

with large wetting contact angles, such as $MgO_x:Ag$, $SiO_xN_y:Ag$ and $HfO_x:Ag$ (Supplementary Ref. S2), may be considered. It is reported that the conductive filament may quickly shrink to original Ag nanoclusters driven by interfacial energy in these material systems when the external biasing is removed, so that the reusability of system can be further improved.”

In order to address the reconfigurability issue clearly, we have added Supplementary **Figure S23 and S24**, along with the following sentences in **Page 48** of the revised manuscript “*A reconfigurable computing system is needed for solving different problems with the same structure. In fact, the MAEN system possesses potential for reconfigurability. Fig. S23 gives an example to show that different problems can be solved with the same device structure. We can see that under two different voltage bias schemes, Ag nanoclusters spontaneously align into a connective filament along the optimal path, leading to two different connectivity patterns. In the case of Fig. S23a, V and 0 were applied to the neighboring terminals T_1 and T_4 respectively, while the other two terminals (i.e. T_2 and T_3) were biased at $1/2V$. In this case, both distance modulation and voltage modulation are involved. The voltage biases V and $1/2V$ were applied to the paths with equal length “ $T_1 \leftrightarrow i_3 \leftrightarrow T_4$ ” and “ $T_3 \leftrightarrow i_2 \leftrightarrow T_4$ ”, constituting the voltage modulation mode. Since the applied voltage bias between T_1 and T_4 is higher, the conductive filament was formed along the path “ $T_1 \leftrightarrow i_3 \leftrightarrow T_4$ ” where the electric field is highest, leading to a significant increase of current from the terminals T_1 and T_4 . In another case of Fig. S23b, V and 0 were applied to the diagonal terminals T_1 and T_3 respectively, while the other two terminals (i.e. T_2 and T_4) were assigned to $1/2V$. Compared with the previous case, a different pattern connecting the terminals T_1 and T_3 by way of the metal islands i_2 and i_3 (i.e. “ $T_1 \leftrightarrow i_3 \leftrightarrow i_2 \leftrightarrow T_3$ ”) was formed. In addition, the graph problem shown in Fig. 3e can also be solved by using the device structure shown in Fig. S23, where the original two-terminal device structure (Fig. 3f) is included as a sub-graph. In future research works, we propose that the back gates can be further added to achieve gate modulation. Once the back gates are incorporated, the weight can be represented by the gate control signal, which provides another degree of freedom to effectively regulate cluster*

evolution and filament growth. By applying a driving bias signal while the control signals representing weight information are applied to the corresponding gate. The gate modulation scheme combines the advantages of distance modulation and voltage modulation schemes, simultaneously possessing high efficiency, high flexibility and high reconfigurability for solution of problems. **Fig. S24** gives an example of solving a 3×3 maze problem based on the gate modulation unit. Thanks to the same device structure independent of the specific maze, a system mapping the $N \times N$ maze is capable of solving arbitrary $n \times n$ maze problems ($n \leq N$), suggesting high reconfigurability of the system.”

In addition, we have included the following discussion in **Page 11** of the revised manuscript “*It is worthwhile noting that reusable and reconfigurable computing units are desirable, and thus the computational costs can be effectively saved. As a proof of concept, our experiments have demonstrated the reusability (Supplementary Fig. 22) and reconfigurability (Supplementary Fig. 23-24) of the MAEN system for the solution of problems, and the detailed discussions can be seen in the Supplementary Information.*”

2. Though the time complexity of Ag nanocluster reorganization is $O(1)$, the time that is needed to fabricate the structures and to visualize the optimal path is tremendous compared to conventional computers. And the advantage of this hardware system would be significantly reduced in this case. Thus, can the authors discuss the potential of solving this problem?

Our response: We would like to thank the reviewer for raising the question. In this work, the solution result represented by the connectivity pattern of the filament was mainly obtained by SEM observations, and the time-dependent current measurement was used as an auxiliary readout method, since the detailed connectivity pattern may not be completely reflected through the limited number of probes in the testing probe stations. The reviewer is also correct that the time spent to fabricate the structures for solving different problems and to visualize the optimal path for obtaining the final

results will reduce the inherent advantage of the system in time complexity. This problem may be addressed in several aspects in the future:

1) the reusability and reconfigurability of the MAEN system, as discussed in the response to the reviewer's previous question, can effectively save the overhead caused by the device fabrication for solving different problems;

2) a dedicated circuit platform for electrical measurements as depicted in **Fig. S30** of the revised manuscript can be developed to support efficient reading and writing of the MAEN device, and in this case the computational efficiency will be significantly improved compared with SEM observation. We should point out that the SEM observation in this work is to demonstrate the correct solution of the problems using MAEN, and in the meantime the gap distance and compliance current were still relatively large to ensure sufficiently long and thick filament(s) for clear observation. Once the SEM observation is replaced by reading and writing periphery, both the gap distance and compliance current can be further reduced, and these will also contribute to the reduction of the solution time and therefore further enhancing the computational efficiency.

Figure S30 | Schematic diagram of circuit platform including writing and reading periphery for MAEN. The module “CTRL” is the controller that sends control signals to decide corresponding operations, such as the terminals to be opened and the voltage bias to be applied. The multiplexer “MUX” receives control signal from “CTRL” and selects the voltage bias that corresponds to the terminals. The signal converter “ADC & DAC” is used to input and read the analog electrical signal.

In order to clarify this point, we have added **Figure S30** as well as related discussions in **Page 54** of the revised manuscript “*In this work, the solution result represented by the connectivity pattern of the filament was mainly obtained by SEM observations, and the time-dependent current measurement was used as an auxiliary readout method, since the detailed connectivity pattern may not be completely reflected through the limited number of probes in the testing probe stations. To achieve efficient writing/reading when the problems to be solved contain multiple inputs/outputs, a dedicated circuit platform for electrical measurements can be developed for the MAEN system. Fig. S30 depicts a schematic diagram of the reading and writing periphery circuits. We should point out that the SEM observation in this work is to demonstrate the correct solution of the problems using MAEN, and in the meantime the gap distance and compliance current were still relatively large to ensure sufficiently long and thick filament(s) for clear observation. Once the SEM observation is replaced by reading and writing periphery, both the gap distance and compliance current can be further reduced, and these will also contribute to the reduction of the solution time and therefore further enhancing the computational efficiency.*”

and in **Page 14** “*It is worthwhile noting that the solution result represented by the connectivity pattern of the filament was mainly obtained by SEM observations, and the time-dependent current measurement was used as an auxiliary readout method, since the detailed connectivity pattern may not be completely reflected through the limited number of probes in the testing probe stations. The reusability and reconfigurability of the MAEN system, as discussed above, can effectively save the overhead caused by the device fabrication for solving different problems, and a dedicated circuit platform including reading and writing periphery (Supplementary Fig. 30) can be developed to probe the MAEN result. In this case, both the gap distance and compliance current can be further reduced, which will also contribute to the reduction of the solution time and therefore further enhancing the computational efficiency.*”

3. In Figure 3, the electrodes with voltage applied on always have sharp tips while

the metal island in the middle are always circular pads. This seems to reduce the generality of the hardware system. If all the electrode pads of a hardware system share the same shape, then the system will be able to compute the optimal route between each two points, and the capability of the whole system will be much more promising.

Our response: We would like to thank the reviewer for raising the question and giving the valuable suggestions. In this manuscript, the tip of electrodes terminals is always sharp to concentrate the electric field, while the shape of metal islands is always circular to achieve uniform regulation on the surrounding electric field in all directions. In order to demonstrate the potential of system generality, the sharp tip of electrode terminals is replaced by more rounded shape to share a similar geometry with the circular metal islands. From **Fig. S25** one can see that the connectivity pattern of conductive filaments after changing the terminal shape (**Fig. S25b**) is exactly the same as before (**Fig. S25a**) under the same voltage scheme. Two shortest paths were simultaneously founded relying on the self-organized evolution of Ag nanoclusters. The results indicate more general and promising application scenarios of the MAEN devices.

Figure S25 | The impact of terminal shape on the connectivity pattern under the same voltage bias scheme. a, SEM image of connection path in the device structure with sharp terminal tips and circular metal islands under the voltage bias scheme. The applied voltage V was 30 V. Scale bar: 200 nm. **b**, SEM image of connection path in the device structure where the sharp terminal tips in (a) were replaced by the more rounded terminal tips. The applied voltage V was 30 V. Scale bar: 200 nm. Two shortest

paths were founded in (a) and (b) relying on the self-organized evolution of Ag nanoclusters.

In order to clarify this point, Supplementary **Figure S25** and related discussion have been included in **Page 50** of the revised manuscript “*In order to demonstrate the potential of system generality, the sharp tip of electrode terminals was replaced by more rounded shape to share a similar geometry with the circular metal islands. From Fig. S25 one can see that the connectivity pattern of conductive filaments after changing the terminal shape (Fig. S25b) is exactly same as before (Fig. S25a) under the same voltage scheme. Two shortest paths were simultaneously founded relying on the self-organized evolution of Ag nanoclusters. The result indicates that the optimal route between each two nodes in the MAEN system may be computed because they share similar geometry, providing more general and promising application scenarios.*”

In addition, the following sentence has been added in **Page 11** of the revised manuscript “*Here, the tip of electrode terminals is always sharp to concentrate the electric field, while the metal islands are always circular to achieve uniform regulation on the surrounding electric field in all directions. In order to demonstrate the potential of system generality, the sharp tip of electrode terminals can be replaced by more rounded shape to share a similar geometry with the circular metal islands, and the same connectivity patterns can be achieved independent of the electrode geometry, indicating more generalized application (see more detailed discussions in Supplementary Fig. 25, Supporting Information).*”

4. It is shown in Figure 1 that the basic modulation units can be controlled by both voltage amplitude and electrode distance. However, only electrode distance modulations are used within the following solution of graph problems and artificial potential field. Can the authors show how voltage amplitude modulation can play a role during solving the problems?

Our response: We would like to thank the reviewer for raising the question. In fact, the distance modulation unit and voltage modulation unit, as two basic modulation schemes, can represent different input information when mapping the problems. For

example, cost and reward need to be comprehensively considered for obtaining the optimal decision in many practical problems. Gap distance reflects the length of path that conductive filament needs to connect, and thus it can be used to express the cost. As the driving signal for the evolution of Ag nanoclusters, different voltage bias schemes applied to the terminals can represent different levels of reward. In the hybrid distance-voltage modulation unit combining two basic modulation schemes, gap distance and voltage bias will jointly modulate the cluster evolution, and the connectivity pattern of conductive filament shown in **Fig. 10-u** represents the optimal decision after comprehensively considering the cost and reward. By using the hybrid modulation scheme, different input information can be directly integrated in the MAEN, thus reducing the overhead required for information processing.

In fact, the two basic modulation schemes have their own advantages and disadvantages in terms of mapping and solution of the problems. For the distance modulation scheme, the graph information is directly mapped with the device structure, and the weights of edges are represented by regulating the positions of electrode terminals or metal islands. The advantages and disadvantages include: 1) When solving the problems, the same voltage biases are applied to the terminals corresponding to the same node, so that these terminals can be connected to a common electrode with constant bias for saving area overhead; 2) Since the weights of edges are directly reflected in the device structure, the problems can be conveniently and efficiently solved in one step by applying voltage biases between the electrodes representing the specified nodes while the intermediate nodes remain floating; 3) The disadvantage is different device structures need to be used to map different graph information, resulting in poor reconfigurability.

In contrast, in case of the voltage modulation scheme the weights are represented by the voltage biases applied to the electrode terminals. The advantages and disadvantages include: 1) The same device structure can be used to represent different graph problems, hence leading to higher reconfigurability; 2) Since each weight needs to be represented by the voltage bias applied to the corresponding electrode, the device will occupy a large area when the number of edges is large.

The distance modulation was used during the solution of graph problems and artificial potential field problem, but in fact voltage modulation can also be involved in solving these problems. **Fig. R3** illustrates a typical example. Under the voltage bias scheme shown in **Fig. R3**, the voltage biases V and $1/2V$ were applied to the paths with equal length “ $T_1 \leftrightarrow i_3 \leftrightarrow T_4$ ” and “ $T_3 \leftrightarrow i_2 \leftrightarrow T_4$ ”, respectively, constituting the voltage modulation mode. Since the applied voltage bias between T_1 and T_4 is higher, the conductive filament was formed along the path “ $T_1 \leftrightarrow i_3 \leftrightarrow T_4$ ” where the electric field is highest, leading to a significant increase of current from the terminals T_1 and T_4 , as can be seen in Fig. R3c. Therefore, to meet the requirements in different application scenarios, an optimal modulation scheme can be designed to improve the efficiency and flexibility for problem solution by tuning gap distance d and voltage bias V separately or collectively based on the respective characteristics of the two modulation units.

Figure R3 | Demonstration of voltage modulation scheme in the device with 4 terminals (T_1, T_2, T_3 and T_4) and 3 metal islands (i_1, i_2 and i_3) for problem solving. **a**, SEM image of device morphology before memristive switching. scale bar: 200 nm. **b**, SEM image of device in (a) after electrical stimulation. scale bar: 200 nm. V and 0 were respectively applied to the neighboring terminals T_1 and T_4 , while the other two terminals (i.e. T_2 and T_3) were biased at $1/2V$. The conductive filament finally established connection between the terminals T_1 and T_4 by way of metal island i_3 . **c**, Corresponding time-dependent current measurement from 4 terminals. The voltage bias V was 15 V.

In order to clarify this point, we have added the following sentences in **Page 8** of the revised manuscript “*The distance and voltage modulation units as two fundamental*

building blocks of MAEN system have their own advantages and disadvantages in terms of mapping and solution of the problems. For the distance modulation unit, the graph information is directly mapped with the device structure, which facilitates efficient solution of problems and saves overhead in electrode area, but leads to poor reconfigurability since the weights are represented in hardware. On the contrary, for the voltage modulation unit, the weights are represented by the voltage biases applied to the electrode terminals, so that the same device structure can be used to represent different graph problems, hence leading to higher reconfigurability. To meet the requirements in different application scenarios, an optimal modulation scheme can be designed to improve the efficiency and flexibility for problem solution by tuning gap distance d and voltage bias V separately or collectively based on the respective characteristics of the two modulation units. Under the hybrid distance-voltage modulation scheme, different input information can be directly integrated in the MAEN, thus enhancing the computing efficiency.”

and in **Page 48** *“In the case of **Fig. S23a**, V and 0 were applied to the neighboring terminals T_1 and T_4 respectively, while the other two terminals (i.e. T_2 and T_3) were biased at $1/2V$. In this case, both distance modulation and voltage modulation are involved. The voltage biases V and $1/2V$ were applied to the paths with equal length “ $T_1 \leftrightarrow i_3 \leftrightarrow T_4$ ” and “ $T_3 \leftrightarrow i_2 \leftrightarrow T_4$ ”, constituting the voltage modulation mode. Since the applied voltage bias between T_1 and T_4 is higher, the conductive filament was formed along the path “ $T_1 \leftrightarrow i_3 \leftrightarrow T_4$ ” where the electric field is highest, leading to a significant increase of current from the terminals T_1 and T_4 .”*

Reviewer #3 (Remarks to the Author):

Overall Remarks: The paper deals with a network based on self-organizing Ag nanoclusters and the use of induced electrical paths as optimized solutions of graph problems and gradient descent. It is well written, interesting and complete, also considering the huge supplementary material. Furthermore, it is timely, since the field of designless computing with similar networks based on nanowires,

nanoparticles or random dopants is becoming much popular every day.

I have two main requests that I think should be satisfied before proceeding with the publication:

Our response: We would like to sincerely thank the reviewer for the positive evaluation and valuable suggestions. We have carefully addressed all the points, as shown below.

1. Although being an interesting and complete paper, no comparison at all is done with current literature employing similar networks based on nanowires, nanoparticles or random dopants. I think a deep discussion should be done versus works of Brown, van der Wiel and Gimzewski (at least).

Our response: We would like to thank the reviewer for the valuable suggestion. In light of the reviewer's advice, we have added detailed discussion on existing studies of nanowires, nanoparticles or random dopants based networks and the novelty of our study in **Page 3** of the revised manuscript:

“In recent years, physical networks with nanoarchitecture composed of nanowires²⁰⁻²⁴, nanoparticles²⁵⁻²⁷ or random dopants²⁸ have been exploited to efficiently implement complex computational tasks in materia based on the nonlinear interactions of the individuals. Gimzewski et al designed atomic switch networks (ASN) composed of multiple overlapping Ag nanowire junctions. The distributed spatiotemporal dynamics of ASN shows great potential for the efficient implementation of reservoir computing^{20,21}. Brown et al studied the avalanches and self-organized criticality originated from spatiotemporal correlations in percolating nanoparticle network. The statistical distributions of avalanche in the percolating nanoparticle network exhibit qualitative and quantitative similarity to those measured in the cortex, which provides a novel architecture for efficient brain-like computing^{26,27}. van der Wiel et al exploited the nonlinearity and tunability of hopping conduction in silicon-based network of boron dopant atoms, enabling efficient implementation of machine learning tasks such as classification²⁸. The rich and complex dynamics of these physical networks have shown great potential in unconventional computing with high energy efficiency.

Here, we report a nanoscale, solid-state multi-agent evolutionary network (MAEN) based on self-organization of distributed Ag nanoclusters. Different from the existing physical networks with immobile elements, here the Ag nanoclusters in the MAEN system as de-centralized agents exhibit spontaneous dynamic evolutions via field-driven ion migrations and electrochemical reactions. The positive feedback by the alignment of Ag nanoclusters leads to the cooperation and competition between the agents, and the resultant connectivity pattern conforms to the principle of optimization. The self-organized evolution of Ag nanoclusters with positive feedback has commonalities in principle with the swarm intelligence, such as the foraging process of ant colony. The kinetic factors in this process, including electric field and ion mobility, provide effective means to the modulation of evolution dynamics.”

2. Even if it is claimed just as an “encouraging pathway toward energy efficient computing hardware”, a discussion on power consumption is needed. Honestly, it is convincing that optimal paths are created, but considering applied voltage (tens of V), timescale (tens of seconds) and measured current (hundreds of nA to several microamps), the power consumption looks huge (from micro to even milli Joule?). How could it be reduced, to follow the above-mentioned claim?

Our response: We would like to thank the reviewer for raising the question. The power consumption of the MAEN device could be optimized in several aspects:

1) We should point out that the SEM observation in this work is to demonstrate the correct solution of the problems using MAEN, and hence the gap distance (d) and compliance current (I) were still relatively large to ensure sufficiently long and thick filament(s) for clear observation. In the future, a dedicated circuit platform including reading and writing periphery can be developed to probe the MAEN result (such as that depicted in Supplementary Fig. 30). Once the SEM observation is replaced by reading and writing periphery, both the gap distance (d) and compliance current (I) can be further reduced, and this will also significantly reduce the applied voltage (V) and switching timescale (t). The reduction in compliance current (I), applied voltage (V) and switching time (t) will significantly reduce the power consumption of the MAEN

device.

2) The device fabrication processes can also be optimized to further reduce the power consumption. For example, dielectric materials with higher ion mobility can be used to effectively promote Ag movement, which therefore is able to further reduce the applied voltage (V) and switching time (t).

These optimizations are expected to be capable of dramatically reducing the power consumption of the MAEN devices. In the present work, the diameter of Ag nanoclusters is ~ 10 nm and the gap distance (d) is usually hundreds of nanometers. From **Equation (1)** in the manuscript, one can see that the forming time (t) decreases exponentially with applied voltage bias (V) and increases with gap distance (d). The fitted surface shown in **Fig. 1f** can be expressed as;

$$t = a \cdot e^{-bV/k_B T d} \quad (1)$$

At the room temperature, $k_B T \approx 0.026$ eV and $a = 2891$, $b = 4.245 \times 10^{-9}$. If we assume the voltage bias $V = 2$ V and the gap distance $d = 20$ nm, the calculated forming time in Equation (1) is 2.346×10^{-4} s. Assuming that the current is decreased to several nanoamperes, the order of energy consumption in the MAEN device can be reduced to the tens of pJ (several of volts \times several nanoamperes \times hundreds of microseconds), implying the potential for the optimization of energy consumption.

3) It is worth mentioning that the energy consumption obtained by the above estimation method by directly multiplying the compliance current, applied voltage and forming time is much higher than the actual value. The accurate energy consumption should be calculated by the time integral:

$$E = \int V \cdot I \cdot dt \quad (2)$$

During the forming process, the current is always at a low level (from femtoamp to picoamp) before the conductive filament establishes a connection between the terminals, and only reaches the compliance current after filament formation. Therefore, the actual energy consumption should be much lower than the estimated value above by directly multiplying the compliance current, applied voltage and forming time. As a result, the

actual energy consumption should be even lower, and in consideration of the physics-empowered parallel computing nature of the MAEN system, it once again implies the potential for enhanced energy efficiency.

In order to clarify this point, we have included the following discussions in **Page 55** of revised manuscript “*In this work, relatively large gap distance (hundreds of nanometers) and high compliance current (hundreds of nanoamps to several microamps) were adopted to ensure obvious filament morphology under SEM observations. A rough estimation by directly multiplying the compliance current, applied voltage and forming time leads to relatively high energy consumption from micro to milli Joule. However, it is worth mentioning that during the forming process, the current is always at a low level (from femtoamp to picoamp) before the conductive filament establishes a connection between the terminals, and only reaches the compliance current after filament formation. Therefore, the actual energy consumption should be calculated by the time integral and should be much lower than the estimated value above, namely,*

$$E = \int V \cdot I \cdot dt \quad (S2)$$

In the future, a dedicated circuit platform including reading and writing periphery can be developed to probe the MAEN result. Once the SEM observation is replaced by reading and writing periphery, both the gap distance (d) and compliance current (I) can be further reduced, and this will also significantly reduce the applied voltage (V) and switching timescale (t). The reduction in compliance current (I), applied voltage (V) and switching time (t) will significantly reduce the power consumption of the MAEN device. Furthermore, the device fabrication processes can also be optimized to further reduce the power consumption. For example, dielectric materials with higher ion mobility can be used to effectively promote Ag movement, which therefore is able to further reduce the applied voltage (V) and switching time (t). The above optimizations in gap distance, compliance current and ion transport properties etc. are expected to be capable of dramatically reducing the power consumption of the MAEN devices.”

and in **Page 14** of the revised manuscript “*The optimizations in gap distance, compliance current and ion transport properties etc. are expected to be capable of*

dramatically reducing the power consumption of the MAEN devices. Together with the physics-empowered parallel computing nature of the MAEN system, it implies high potential for enhanced energy efficiency (see more detailed discussion in Supplementary Information).”

Other minor points are:

3. Fig S1 is never cited in the main text

Our response: We would like to thank the reviewer for raising the question.

Supplementary Fig. 1 is now cited in **Page 5** of the main text “*The preparation of the devices is described in Experimental Section and Supplementary Fig. 1*”.

4. Fig S3 is claimed to show the path from AFM, but honestly I don’t see it: what should I see in the pictures?

Our response: We would like to thank the reviewer for raising the question. In order to show the connective path more clearly, we have added some dashed lines as guides for the eyes, together with the corresponding SEM image of the same conductive filament, as shown in **Figure S3**, which is also appended below for the reviewer’s convenience.

Figure S3 | AFM characterization of the micro-topography of device surface. SEM images of two-terminal device **a**, before and **b**, after electrical stimulation, which have been shown in Fig. 1c and 1d, respectively. Scale bar: 100 nm. **c**, AFM image of two-terminal device before memristive switching, corresponding to the SEM image in **(a)**. Scale bar: 100 nm. **d**, AFM image of two-terminal device after applying the voltage bias V , corresponding to the SEM image in **(b)**. The conductive filament shown in **(d)** is marked with dashed lines as guides for the eyes. Scale bar: 100 nm. The applied voltage V was 25 V.

5. Memristive behavior is just shown in some I - V plots in Fig S7. I think some data about endurance and retention would be needed, since from that picture the I - V plots look not that repeatable.

Our response: We would like to thank the reviewer for raising the question. In order to demonstrate the repeatable switching of our system, we performed cycling I - V sweep on the two-terminal device with 100 nm gap. **Fig. S8** shows the repeatable I - V curves under 400 alternating sweep cycles, indicating our devices can be repeatably operated.

Figure S8 | Repeatable I - V curves of two-terminal device with 100 nm gap under 400 alternating sweep cycles. The device can be repeatably switched under the alternating positive and negative voltage sweeps.

From the I - V curves one can see that the device shows threshold switching characteristics due to the spontaneous dissolution of conductive filament driven by the interfacial energy minimization (Ref. R1). Therefore, the device will relax back to the off state after removing the electrical stimulation. Although the continuous filament is broken into discrete clusters, they do not fully recover to their original positions. As a result, the threshold voltages in the subsequent switching processes are significantly reduced, and a new filament can be easily connected based on the previously incompletely ruptured filament (See **Fig. S9**). For example, when a small voltage bias (2 V) was applied onto a pristine MAEN device with 100 nm gap, there was no switching within 20 s (**Fig. S9a**). Once a large voltage bias (10 V) was applied, a conductive filament can be formed between the two terminals, along with an obvious increase in current at $t \approx 2.5$ s (**Fig. S9b**). Although the device spontaneously returned to off state after removing the bias, the application of a small voltage bias of 2 V has switched the device to on state again within 1 s (**Fig. S9c**), since the incompletely ruptured filament can be recovered by a smaller driving force. The device can maintain the on-state for a long time at this small voltage bias, i.e. >500 s in Fig. S9c.

Figure S9 | The electrical memory effect due to the incomplete rupture of conductive filament. **a**, Time-dependent current measurement when a small voltage bias (2 V) was applied on the two-terminal device with 100 nm gap. There is no obvious change for the current within 20 s. **b**, Time-dependent current measurement on the device in (a) under a larger voltage bias of 10 V for the forming process. The current jumped at $t \approx 2.5$ s, indicating a connection of conductive filament between the terminals. **c**, Time-dependent current measurement under a small voltage bias (2 V) after forming process in (b). The filament was recovered rapidly with 1 s, and the device can maintain its on-state for a long time under this small voltage bias.

In order to show the endurance and retention of MAEN system, we have added **Figure S8, Figure S9** and the related discussions in **Page 34** of the revised manuscript “*The repeatable I-V curves under 400 alternating sweep cycles was given in Fig. S8, indicating the MAEN devices can be stably operated ... The MAEN device shows threshold switching characteristics due to the spontaneous dissolution of conductive filament driven by the interfacial energy minimization. Therefore, the device will relax back to the off state after removing the electrical stimulation. Although the continuous filament is broken into discrete clusters, they do not fully recover to their original positions. As a result, the threshold voltages in the subsequent switching processes are significantly reduced, and a new filament can be easily connected based on the previously incompletely ruptured filament. For example, when a small voltage bias (2 V) was applied onto a pristine MAEN device with 100 nm gap, there was no switching within 20 s (Fig. S9a). Once a large voltage bias (10 V) was applied, a conductive filament can be formed between the two terminals, along with an obvious increase in current at $t \approx 2.5$ s (Fig. S9b). Although the device spontaneously returned to off state after removing the bias, the application of a small voltage bias of 2 V has switched the device to on state again within 1 s (Fig. S9c), since the incompletely ruptured filament can be recovered by a smaller driving force. The device can maintain the on-state for a long time at this small voltage bias, i.e. >500 s in Fig. S9c.*”

and in **Page 6** “*The morphology of conductive filament can be tuned by the compliance current³¹ (Supplementary Fig. 6), and pinched hysteresis loops were observed during I-V measurements and verified the memristive effect³⁶⁻⁴¹ of the two-terminal MAEN devices (Supplementary Fig. 7-9).*”

6. Is there any (short) memory effect? If yes, can it be used for computing?

Our response: We would like to thank the reviewer for raising the question. The pinched hysteresis loops from the repeatable I-V sweep measurements shown in **Fig. S7** confirm the memristive effect of the MAEN devices, since the pinched hysteresis loop is the feature of memristors (Ref. R2). The volatile threshold switching in **Fig. S7**

indicates short-term memory, which can be used for a number of computing applications. For example, the self-organized evolution with nonlinear dynamics and short-term memory properties has been shown to intelligently integrate spatio-temporal information, showing significant advantage in the implementation of in materia reservoir computing (Ref. R3-R5). Therefore, we expect the MAEN system with short-term memory holds similar potential in reservoir computing. However, instead of utilizing its short-term memory effect, in the present study we use the Ag nanoclusters in the MAEN system as de-centralized agents based on their spontaneous evolution dynamics via field-driven ion migrations and electrochemical reactions. The positive feedback by the alignment of Ag nanoclusters leads to the cooperation and competition between the agents, and the resultant connectivity pattern conforms to the principle of optimization. Although this represents different strategy in achieving physics-based computing compared with short-term memory based applications, the two critical components for reservoir computing, i.e. short-term memory and nonlinear dynamics (Ref. R5-R6), are present in the MAEN system, which should allow it to function as reservoir computing system as well.

In order to clarify this point, we have included the following sentences in **Page 6** of the revised manuscript “*The MAEN system with nonlinear dynamics and fading memory property holds potential in efficient implementation of neuromorphic computational tasks, such as reservoir computing^{20,21,24}.*”

7. How the simultaneously writing/reading would be guaranteed in the perspective of several more inputs/outputs?

Our response: We would like to thank the reviewer for raising this important question. As the scale of the problem increases, the number of inputs/outputs may also increase accordingly. In this case, a dedicated circuit platform including writing and reading periphery can be developed to support efficient operation of the MAEN device, as depicted in **Fig. S30** of the revised manuscript. The module “CTRL” is the controller that sends control signals to decide corresponding operations, such as the terminals to be opened and the voltage bias to be applied. The multiplexer “MUX” receives control

signal from “CTRL” and selects the voltage bias that corresponds to the terminals. The signal converter “ADC & DAC” is used to input and read the analog electrical signal. Once the SEM observation is replaced by reading and writing periphery, both the gap distance and compliance current can be further reduced, and these will also contribute to the reduction of the solution time and therefore further enhancing the computational efficiency.

In order to clarify this point, we have added **Figure S30** in the revised manuscript, along with the following discussion in **Page 54** *“To achieve efficient writing/reading when the problems to be solved contain multiple inputs/outputs, a dedicated circuit platform for electrical measurements can be developed for the MAEN system. **Fig. S30** depicts a schematic diagram of the reading and writing periphery circuits. The module “CTRL” is the controller that sends control signals to decide corresponding operations, such as the terminals to be addressed and the voltage bias to be applied. The multiplexer “MUX” receives control signal from “CTRL” and selects the voltage bias that corresponds to the terminals. The signal converter “ADC & DAC” is used to input and read the analog electrical signal.”*

Figure S30 | Schematic diagram of basic MAEN circuit modules for electrical measurement. The module “CTRL” is the controller which sends control signals to decide corresponding operations, such as the terminals to be addressed and the voltage bias to be applied. The multiplexer “MUX” receives control signal from “CTRL” and selects the voltage bias that corresponds to the terminals. The signal converter “ADC & DAC” is used to input and read the analog electrical signal.

8. It is not clear to me why both V amplitude and distance are used: since driving force is the applied electric field, there's no difference in principle in using one or the other, but tuning V is far easier than changing distance by lithography. Please clarify. The metaphor of cost (distance) and reward (voltage) looks obscure to me, since here the two variables are not really independent in forming E .

Our response: We would like to thank the reviewer for raising the question. In our work, the distance modulation unit and voltage modulation unit are proposed to form the two building blocks for MAEN. Although there is no difference in principle where the cluster evolution is driven by the electric field, the two basic modulation schemes have their own advantages and disadvantages in terms of mapping and solution of the problems.

For the distance modulation scheme, the graph information is directly mapped with the device structure, and the weights of edges are represented by regulating the positions of electrode terminals or metal islands. The advantages and disadvantages include: 1) When solving the problems, the same voltage biases are applied to the terminals corresponding to the same node, so that these terminals can be connected to a common electrode with constant bias for saving area overhead; 2) Since the weights of edges are directly reflected in the device structure, the problems can be conveniently and efficiently solved in one step by applying voltage biases between the electrodes representing the specified nodes while the intermediate nodes remain floating; 3) The disadvantage is different device structures need to be used to map different graph information, resulting in poor reconfigurability.

In case of the voltage modulation scheme, the weights are represented by the voltage biases applied to the electrode terminals. The advantages and disadvantages include: 1) The same device structure can be used to represent different graph problems, hence leading to higher reconfigurability; 2) Since each weight needs to be represented by the voltage bias applied to the corresponding electrode, the device will occupy a large area when the number of edges is large.

To meet the requirements in different application scenarios, an optimal modulation

scheme can be designed to improve the efficiency and flexibility for problem solution by tuning gap distance d and voltage bias V separately or collectively based on the respective characteristics of the two modulation units.

To clarify the advantages and disadvantages of the two basic modulation units, the following sentences have been added into **Page 8** of the revised manuscript “*The distance and voltage modulation units as two fundamental building blocks of MAEN system have their own advantages and disadvantages in terms of mapping and solution of the problems. For the distance modulation unit, the graph information is directly mapped with the device structure, which facilitates efficient solution of problems and saves overhead in electrode area, but leads to poor reconfigurability since the weights are represented in hardware. On the contrary, for the voltage modulation unit, the weights are represented by the voltage biases applied to the electrode terminals, so that the same device structure can be used to represent different graph problems, hence leading to higher reconfigurability. To meet the requirements in different application scenarios, an optimal modulation scheme can be designed to improve the efficiency and flexibility for problem solution by tuning gap distance d and voltage bias V separately or collectively based on the respective characteristics of the two modulation units. Under the hybrid distance-voltage modulation scheme, different input information can be directly integrated in the MAEN, thus enhancing the computing efficiency.*”

Reference

- R1. Wang, Z. et al. Memristors with diffusive dynamics as synaptic emulators for neuromorphic computing. *Nat Mater.* **16**, 101-108 (2017).
- R2. Chua, L. If it's pinched it's a memristor. *Semicond. Sci. Technol.* **29** (2014).
- R3. Sillin, H. O. et al. A theoretical and experimental study of neuromorphic atomic switch networks for reservoir computing. *Nanotechnology* **24**, 384004 (2013).
- R4. Demis, E. C. et al. Nanoarchitectonic atomic switch networks for unconventional computing. *Jpn. J. Appl. Phys.* **55**, 1102B2 (2016).
- R5. Milano, G. et al. In materia reservoir computing with a fully memristive architecture based on self-organizing nanowire networks. *Nat. Mater.* **21**, 195-202

(2021).

R6. Torrejon, J. et al. Neuromorphic computing with nanoscale spintronic oscillators.

Nature **547**, 428-431 (2017).

REVIEWERS' COMMENTS

Reviewer #1 (Remarks to the Author):

I thank the authors for the very detailed responses to my questions, which have well addressed my early concerns and improved the clarity of the manuscript. I thus recommend it for publication as is.

Reviewer #2 (Remarks to the Author):

The authors have addressed all my previous concerns in this revision, which might be accepted as is now.

Reviewer #3 (Remarks to the Author):

The authors carefully replied to my concerns. I'm fully satisfied about them. I think the paper is now substantially improved and ready for publication.